# HOW TRANSFORMERS LEARN STRUCTURED DATA: INSIGHTS FROM HIERARCHICAL FILTERING

## ABSTRACT

Understanding the learning process and the embedded computation in transformers is becoming a central goal for the development of interpretable AI. In the present study, we introduce a hierarchical filtering procedure for generative models of sequences on trees, allowing us to hand-tune the range of positional correlations in the data. Leveraging this controlled setting, we provide evidence that vanilla encoder-only transformers can approximate the exact inference algorithm when trained on root classification and masked language modeling tasks, and study *how* this computation is discovered and implemented. We find that correlations at larger distances, corresponding to increasing layers of the hierarchy, are sequentially included by the network during training. Moreover, by comparing attention maps from models trained with varying degrees of filtering and by probing the different encoder levels, we find clear evidence of a reconstruction of correlations on successive length scales corresponding to the various levels of the hierarchy, which we relate to a plausible implementation of the exact inference algorithm within the same architecture.

## 1 INTRODUCTION

Transformer-based large language models have revolutionized natural language processing, and have notably demonstrated their capacity to perfectly assimilate the grammatical rules of the languages they are trained on. While this evidence shows that transformers can handle and exploit the subtle long-range correlations that emerge in natural language, their inner workings remain largely unclear.

Due to the complexity of the standard transformer architecture (Vaswani et al., 2017), understanding what strategy is precisely implemented via the attention mechanism to solve a given problem has been limited so far to very simple tasks (Weiss et al., 2021; Zhong et al., 2024; Behrens et al., 2024). Nonetheless, significant results have been obtained by studying transformers on simplified models of language known as Context-Free Grammars (CFGs). Through probing of the so-called parsing tree of CFGs, evidence has notably pointed towards transformers trained on predicting masked symbols implementing the optimal dynamic programming algorithm to reconstruct the hidden structure of the grammar, but alas without finding a fully plausible implementation within the architecture (Zhao et al., 2023; Allen-Zhu & Li, 2023). On the other hand, when tasked with reconstructing the most probable parsing tree in the context of probabilistic CFGs, transformers may struggle to match the optimal algorithm if ambiguity is high (Khalighinejad et al., 2023).

Beyond language models, the significance of data structure in machine learning applications is well recognized yet remains poorly understood. CFGs represent a data structure characterized by hierarchical correlations (Mossel, 2016). In general, understanding how standard deep networks can take advantage of this hierarchical structure in their training is an important research question. Towards this objective, simplified hierarchical models of structured data on fixed trees have proved very useful in understanding the effectiveness of Convolutional Neural Networks (CNNs) (Cagnetta et al., 2024), for which there are now formal results supporting the idea that the optimal Belief Propagation (BP) algorithm can be approximately implemented (Mei, 2024). Unfortunately, while the implementation of the hierarchy in CNNs is made quite transparent by the hierarchical structure of their convolutional filters, this is not true for transformers, and one can therefore not straightforwardly transpose this interpretation to other architectures (Cagnetta & Wyart, 2024).

In this work, we present a complementary study to those described above, which allows us to understand further *how* transformers approach optimal inference in a structured data model.

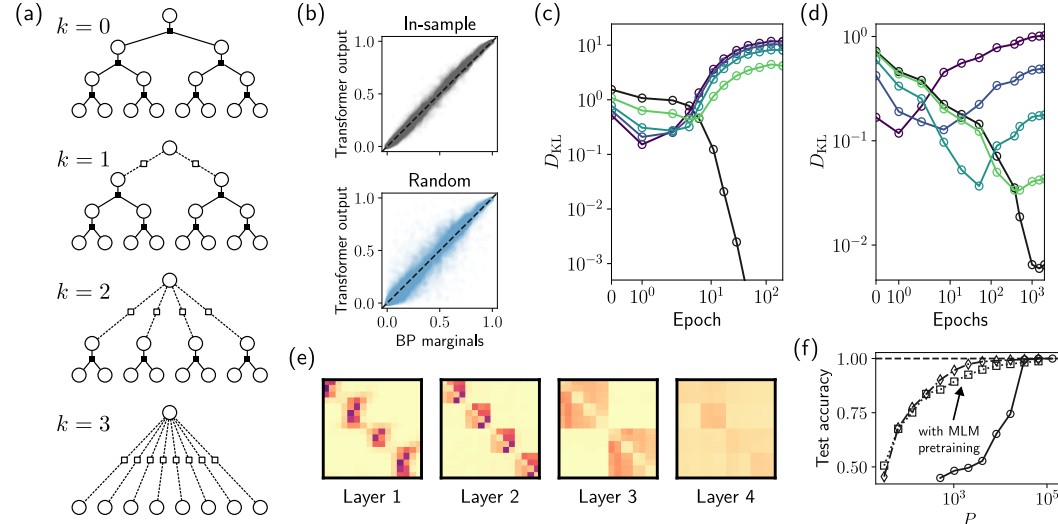

Figure 1: Synthesis of our main results. (a) The proposed filtered hierarchical model, illustrated here with $\ell = 3$ layers and with a filtering parameter $0 \leq k \leq \ell$, allowing one to truncate the hierarchy and generate data with more or less structure. (b) Scatter plot of the predictions of a trained transformer for a masked symbol ($\ell = 4$, $k = 0$, $q = 4$ possible states) versus the corresponding exact marginals obtained with the BP oracle, in-sample on $10^4$ sequences (top), and out-of-sample on uniformly generated sequences (bottom). (c) Evolution along training, on a root classification task with $P = 2^{17}$ examples ($\ell = 4$, $k = 0$, $q = 4$) of the average Kullback-Leibler divergence between transformer predictions and marginals obtained from the matched BP (black) and mismatched BP (from light green $k = 1$ to purple $k = 4$) on identical in-sample inputs, demonstrating the transformer learns increasingly structured representations. (d) Identical to (c) for a MLM task on $P = 2^{18}$ data. (e) Attention maps averaged over $10^4$ in-sample inputs, for a transformer with $n_L = \ell = 4$ layers of attention trained on the MLM task with fully hierarchical data, exhibiting a structure that mirrors the organization of the generative tree and the sequence of operations of BP. (f) Test accuracy on root classification on fully hierarchical data ($\ell = 4$, $k = 0$, $q = 4$) versus number of labeled training samples $P$ with no pretraining (circles) compared to MLM pretraining with frozen (squares) and unfrozen (diamonds) encoder weights during fine-tuning.

**Our contributions.** We propose a controlled hierarchical model of discrete sequences, in which we can easily tune the strength of correlations between tokens thanks to a "filtering" parameter $k$, illustrated in Fig. 1(a). This tree-based probabilistic graphical model gives us access to the *exact* inference algorithm for reconstructing any symbol on the tree, Belief Propagation (BP) (Mézard & Montanari, 2009). Leveraging this context, we show that

- Transformers not only approach optimal performance in root classification and Mask Language Modeling (MLM) tasks, but they spontaneously do so in a calibrated way—i.e., by predicting probabilities that approximate those yielded by the BP oracle even on out-of-sample inputs, see Fig. 1(b)—which provides evidence of an equivalence in computation to the exact inference algorithm.

- When trained with stochastic gradient descent, transformers sequentially discover the existence of higher hierarchical correlation levels (i.e., longer-range correlations), progressively aligning with the prediction of algorithms that impute only parts of the full correlation structure, see Fig. 1(c)-(d). In other words, our simplified setting allows us to understand how transformers learn from structured data in *time*.

- Well-trained transformers reconstruct the correct hierarchical structure through the succession of attention blocks. Matching the number of transformer layers to the number of layers in the generative tree, we find that the attention maps are compatible with a natural implementation of BP within the architecture, see Fig. 1(e). We verify this affinity through probing experiments, providing strong clues on how transformers learn from our structured data in *"space"*, thereby explaining the effectiveness of unsupervised pre-training for supervised classification tasks, illustrated in Fig. 1(f).

The paper is organized as follows. First, we provide a detailed description of our tunable hierarchical model in Sec. 2. We then perform numerical experiments on standard transformer architectures in Sec. 3, shedding light on the learning dynamics. The understanding of the implementation learned by the transformer, and its compatibility with a possible implementation of the Belief Propagation algorithm in the architecture that we propose, is analyzed in-depth in Sec. 4. We finally conclude and discuss the wider implications of our results in Sec. 5.

## 2 A MODEL WITH FILTERED HIERARCHICAL CORRELATIONS

### 2.1 THE FULL HIERARCHICAL MODEL

We consider a tree-based generative process producing structured sequences of discrete symbols. We here focus on the fixed tree topology case, allowing for direct control over the effective range of the hierarchical correlations induced in the generated sequences (2.2), and enabling exact and efficient inference through Belief Propagation (2.4).

The "full" hierarchical generative process shown in the first row of Fig 1(a) can be described as follows. The chain starts from an initial symbol $x_0$, which we will refer to as the *root* of the tree, sampled with probability $\boldsymbol{p}_0$ from a vocabulary $\mathcal{X} = \{1, \ldots, q\}$. Then, the first layer of the tree is drawn randomly using a transition tensor $\mathsf{M}$, which assigns the probability of generating some children—from the same vocabulary $\mathcal{X}$—given a parent (here $x_0$). In this work, we will restrict ourselves to binary trees for simplicity. We therefore have $\mathsf{M} \in \mathbb{R}_+^{q \times q \times q}$, with $M_{abc}$ the probability of generating the *pair* $(b, c)$ given a parent $a$. Since its elements are transition probabilities, this tensor should satisfy $M_{abc} \in [0, 1] \ \forall \, a, b, c$ and $\sum_{bc} M_{abc} = 1 \ \forall \, a$. The process, with the same tensor $\mathsf{M}$, is then repeated independently for each of the newly created children nodes for a total of $\ell$ generations, eventually yielding a sequence of $2^\ell$ symbols $\{x_i\}_{i=1,\ldots,2^\ell}$. We will refer to the symbols in the sequence as the *leaves* of the generative tree.

The class of transition tensors $\mathsf{M}$ that we use is defined precisely in Appendix A. In short, we will resort to randomly sampled log-normal transition probabilities, yielding complex long-range correlations along the sequences. Importantly, we will only consider tensors with non-overlapping entries, such that: if $M_{abc} > 0$, then $\forall a' \neq a \ M_{a'bc} = 0$. As a result, the production rules of our unfiltered generative model are *non-ambiguous* in the sense that a pair of children symbols can only have a single parent. Given all the symbols on the leaves, one can therefore *deterministically* reconstruct the underlying generative tree, all the way up to the root.

### 2.2 FILTERING HIERARCHICAL CORRELATIONS

We develop a filtering tool that enables control over the correlation structure in the generated sequences. In particular, we consider a family of generative models, indexed by an integer $k \leq \ell$, with hierarchical correlations truncated at a given depth $k$ of the tree.

In the $k = 0$ case described in the previous paragraph, all children generated at any level of the tree are sampled in pairs from their respective parents and are strongly correlated. When $k > 0$, we instead generate the tree by drawing the children at level $k$ *conditionally independently* given the root, with the same marginals as the full ($k = 0$) model. Then, for layers below layer $k$, the generative process is the standard one described above, inducing correlations within blocks of $2^{\ell-k}$ tokens. The procedure is illustrated in Fig. 1(a), where dashed segments indicate conditional independence.

In order to match the correct marginal probabilities in the truncated models, the conditional independent sampling at level $k$ is done as follows. For each of the $2^k$ variables at level $k$, say $x_j$,[1] one considers the unique path that relates the root to this intermediate child in the original fully hierarchical tree, yielding a probability

$$P\left(x_j = b \mid x_0 = a\right) = \left(\boldsymbol{p}_0 \boldsymbol{M}^{\sigma_0(j)} \boldsymbol{M}^{\sigma_1(j)} \ldots \boldsymbol{M}^{\sigma_{k-1}(j)}\right)_{a,b}, \tag{1}$$

with $\sigma_m(j) \in \{L, R\}$ indicating whether the path leading to the tree element $j$ considered at layer $k$ takes a left or right branching at the previous layer $m$. The $q \times q$ transition matrices $\boldsymbol{M}^L$ and $\boldsymbol{M}^R$

---

[1] Here we take $j > 2^\ell$ to refer to the internal nodes of the tree, while $x_0$ remains the root and $x_i$ with $i = 1, \ldots, 2^\ell$ are the leaves.

are computed by tracing the original tensor

$$M_{ab}^L = \sum_c M_{abc}, \qquad M_{ac}^R = \sum_b M_{abc}, \tag{2}$$

By constructing filtered trees in such a way, we ensure that the conditional correlations of the leaves capture up to the $k^{\text{th}}$ level of the hierarchy. Note, however, that when $k > 0$ the root can no longer be recovered deterministically from the leaves.

## 2.3 RELATED DATA MODELS

**Context-free grammars.** Our hierarchical model can be considered as an instance of a simplified probabilistic context-free grammar (PCFG) with log-normally distributed transition rates (De Giuli, 2019). The simplification is two-fold. Standard CFGs typically include two distinct sets of symbols, non-terminals and terminals, representing parts of speech—i.e. nouns, verbs etc.—and actual words respectively, plus a root symbol. Here, instead, we consider a single vocabulary $\mathcal{X}$ for all the symbols in the tree, including the root—which allows us to define a root classification task. Moreover, the *parsing trees* underlying CFGs are not fixed: terminals can be produced at different levels and the sequence length can vary. Instead, we assume a fixed parsing tree for our model, where the $2^\ell$ leaves are collected from the last layer—which allows us to define a filtering procedure based on removing layers of hidden symbols above the leaves.

**The Random Hierarchy Model.** Our model is closely related to the recently introduced Random Hierarchy Model (RHM) of Cagnetta et al. (2024), which was studied to improve the understanding of the effect of hierarchical structures on generative diffusion (Sclocchi et al., 2024) or last token prediction (Cagnetta & Wyart, 2024). The main differences to our formulation are that in the RHM the allowed transitions have uniform transition rates—while we consider a log-normal distribution—and that the production rules depend on the layer—while we here consider a single transition tensor throughout the tree. Correlations between the leaves arise in the RHM when some children pairs cannot be produced, leading to a reduced entropy of viable sequences. Having non-uniform transitions in our model similarly limits the entropy, while leading to a significantly different correlation structure. One should for instance notice that the staircase decrease of the correlations as a function of the distance between leaves presented in Cagnetta & Wyart (2024) is not visible in our case.

## 2.4 EXACT INFERENCE

A key advantage of generating sequences through a tree-based process is that we can perform exact inference efficiently using a dynamic programming approach. Moreover, the fixed tree topology allows us to consider a simplified version of the general *inside-outside* algorithm (Baker, 1979), which can be written in a message-passing form within the Belief Propagation (BP) formalism (Sato, 2007; Mézard & Montanari, 2009). Assuming that the transition tensor $\mathbf{M}$ and root probabilities $\boldsymbol{p}_0$ are known, with BP one can compute the exact marginal probabilities for all the symbols at any position in the tree, with a computational cost linear in the size of the tree. Without going into detail on the derivation, let us describe the BP scheme for the filtered tree graphs we are considering.

We start by randomly initializing an upgoing and downgoing message—each one being a vector in $\mathbb{R}^q$ that represents a probability distribution over the $q$ possible symbols—for each edge in the generative tree. In the following, we denote with $\nu_{j\to\alpha}$ a message going from a so-called variable node $j$ (shown by a circle in the sketches) to a factor node $\alpha$ (shown by a full or empty square in the sketches), and with $\hat{\nu}_{\alpha\to j}$ the message in the opposite direction. Wherever there is a known variable one should then fix $\nu_{j\to\alpha}[x_j] = \delta_{x_j,a}$, where $a$ is the known value e.g. of the leaf.

When the hierarchy is truncated, two distinct types of updates are possible, depending on whether one lies in the filtered or unfiltered regions of the tree. In the former, the root is directly connected to $2^k$ "empty" factor nodes, as shown in Fig. 2(a), each connected to a single and distinct variable node below. In this case the BP fixed point equations for messages from the root to the empty factor are given by

$$\nu_{0\to\alpha_j}[x_0] \propto \prod_{\ell\neq j} \hat{\nu}_{\alpha_\ell\to 0}[x_0], \tag{3}$$

i.e. outgoing messages are simply a product of the incoming messages from all the other edges. At each of the $2^k$ factor nodes, both upgoing and downgoing messages satisfy

$$\hat{\nu}_{\alpha_j \to 0}[x_0] \propto \sum_{x_j} P(x_j \mid x_0) \nu_{j \to \alpha_j}[x_j], \qquad \hat{\nu}_{\alpha_j \to j}[x_j] \propto \sum_{x_0} P(x_j \mid x_0) \nu_{0 \to \alpha_j}[x_0], \quad (4)$$

where $P(x_j \mid x_0)$ is given by equation 1, and is specific to the factor node considered. The notation $\propto$ means that the messages—that are probabilities—are to be normalized (e.g. $\sum_{x_0} \hat{\nu}_{\alpha_j \to 0}[x_0] = 1$).

We now consider the lower, unfiltered part of the tree. As illustrated in Fig. 2(b), each of the "full" factor nodes is connected to three variable nodes, representing the parent and two children in the standard branching process. The outgoing messages from the factor node should satisfy

$$\hat{\nu}_{\alpha \to u}[x_u] \propto \sum_{x_l, x_r} M_{x_u x_\ell x_r} \nu_{l \to \alpha}[x_l] \nu_{r \to \alpha}[x_r]. \quad (5)$$

For all variable nodes except for the root detailed above, the single outgoing messages are equal to the single incoming messages in these variable nodes at the previous/next layer of the tree. For example, the upgoing messages $\nu_{1 \to \alpha_1}$ in Fig. 2(a) is simply $\hat{\nu}_{\alpha \to 1}$, where $\alpha$ is the *full* factor node lying below variable 1 (assuming $k < \ell$). Efficient convergence to the fixed point is guaranteed if one starts from the leaves and updates the messages in an upgoing pass, and then performs a downgoing pass from the root, for a total of $2(\ell - k + 1)$ steps. Once the messages have converged, any unknown variable can be optimally reconstructed by computing the marginals as

(a)

(b)

Figure 2: Illustration of the two types of BP updates: (a) above; (b) below the filter level $k$.

$$\mu[x_i] \propto \prod_{\alpha \in \partial i} \hat{\nu}_{\alpha \to i}[x_i], \quad (6)$$

where $\partial i$ is the set of factor nodes connected to variable node $i$. In our problem, this product will therefore typically be over a single factor node when inferring masked leaves, or $2^k$ factor nodes when inferring the root.

In the following, we will adopt the short-hand notation $\mathrm{BP}_k$ to denote a BP implementation that assumes the computational graph of the $k$-filtered hierarchical model, thus able to perform exact inference in a matched case with data with filtering parameter equal to $k$.

## 3 How transformers learn to climb the hierarchy in time

### 3.1 Experimental setup

We will focus on the encoder-only variant (Devlin et al., 2019) of the celebrated "vanilla" transformer architecture, introduced in Vaswani et al. (2017). A full recap of this parametrization is given in Appendix B.

In a nutshell, each of the sequence elements $x_i \in \{1, \ldots, q\}$ is first converted to a positionally-informed token $\boldsymbol{x}_i^{(0)} \in \mathbb{R}^d$. For our experiments, we consider $d = 128$ and the standard sinusoidal positional encoding of Vaswani et al. (2017). Each transformer block in the network then maps the previous encoded sequence onto a new sequence of tokens with the same length and embedding dimension, through a concatenation of a self-attention layer and a fully connected layer, with residual connections and layer normalization. The self-attention layer importantly introduces some mixing between the different tokens in the sequence, represented by what we will refer to as an attention matrix $\boldsymbol{A} \in \mathbb{R}_+^{2^\ell \times 2^\ell}$. We take the fully connected layer to be a standard 2-layer network with relu activations and hidden dimension $d' = 2048$. Following these operations, repeated $n_L$ times to obtain the full encoder, we obtain a position-dependent high-dimensional representation of each of the original symbols in the sequence. What is finally done with this sequence of tokens depends on the task at hand: we consider root classification in Sec. 3.2 and masked language modeling in Sec. 3.3.

Motivated by our focus on understanding the transformer's implementation, we will take the number of attention layers to match the depth of the unfiltered generative tree, $n_L = \ell$. Studying varying values of $k$ for the training data will effectively allow us to explore cases where there are more attention layers than hierarchical levels in the generative tree, while we discuss the consequences of having $n_L$ smaller than the number of hierarchical levels in Appendix D.1.

In the following, all numerical experiments are performed on the same realization of the transition tensor, randomly sampled for $q = 4$ using the parametrization described in Appendix A (see also our Reproducibility Statement below). While there may be quantitative differences for different randomly generated tensors—particularly at small $q$—results remain qualitatively unchanged in experiments on different grammars, see Appendix D.2.

## 3.2 SUPERVISED CLASSIFICATION

In the context of our model, a natural idea is to use the root of a tree $x_0$ as a label for the generated sequence $\{x_i\}$, and to train a transformer encoder architecture on the associated classification task using a dataset of $P$ *labeled* sequences. To perform the root prediction, the tokens in the final layer are concatenated position-wise (forming a large $d \times 2^\ell$ vector) and fed to a linear readout, which outputs $q$ logits associated with the possible root symbols. The network is trained by minimizing the cross-entropy loss between these logits and the correct one-hot encoding of the root.

**Optimal test accuracy.** We find that given sufficient labeled data $P \geq P^*$, transformers achieve perfect in-sample root classification accuracy in the fully hierarchical model, $k = 0$, as illustrated in Fig. 3. When the training data has filtering parameter $k > 0$, the networks approach the optimal in-sample accuracy predicted by $\mathrm{BP}_k$, see Fig. 10 of Appendix D.3. Notice that, while in the case $k = 0$ the exact algorithm finds the value of the root with accuracy 1, this is no longer the case for $k \geq 1$ where the optimal accuracy is $< 1$.

Different from the Random Hierarchy Model of Cagnetta et al. (2024), characterizing analytically the scaling of $P^*$ with the parameters of the grammar with our non-uniform transition probabilities is a challenging goal, and is left for future work. Still, we discuss the role of the filtering parameter $k$ of the data model on the sample complexity in Appendix D.3

**Out-of-sample testing.** In our data model, one can also test out-of-sample with respect to the filtering parameter $k$. For example, we test models trained on intermediate filtered data on a fully hierarchical dataset, i.e., $k_{\mathrm{train}} > 0$ and $k_{\mathrm{test}} = 0$, in Fig. 3, or vice-versa, i.e., $k_{\mathrm{train}} = 0$ and $k_{\mathrm{test}} > 0$, in Fig. 4. In both cases, the transformers achieve a performance that exactly matches that of $\mathrm{BP}_{k_{\mathrm{train}}}$, in the presence of the same mismatch between the assumed inference model and the data generative model. We stress that, in this mismatched task, the BP prediction is no longer optimal, yet the trained networks systematically reach the same accuracy. This observation provides the first evidence that the transformers are implementing an approximation of the $\mathrm{BP}_{k_{\mathrm{train}}}$ algorithm matched to the training data distribution.

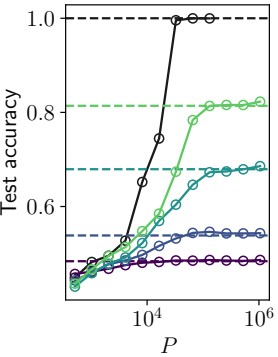

Figure 3: Evolution of the root prediction accuracy on full hierarchical $k_{\mathrm{test}} = 0$ test samples for transformers trained on $P$ labeled samples generated with $k_{\mathrm{train}} = 0, 1, 2, 3, 4$ (top to bottom). Dashed lines indicate, for each $k$, the accuracy computed with the $\mathrm{BP}_k$ algorithm on unfiltered data.

**Full prediction matching.** So far, we have established that the trained transformers match the accuracy of the exact inference algorithm on the root prediction in- and out-of-sample. We can however go one step further, as the transformers output $q$ logits, which were passed through an $\arg\max$ operation to yield a prediction. Taking the $\mathrm{softmax}$ instead gives a normalized $q$-dimensional vector, which we can interpret as the predicted probabilities of the root symbol given the input sequence, to be compared to the *exact* marginals obtained with BP. We find a close match at the end of training, as shown by the small Kullback-Leibler divergences averaged over in-sample inputs in the $k = 0$ case in Fig. 1(a), and similarly for $k \geq 0$, on both in-sample and entirely out-of-sample inputs in Fig. 11 of the Appendix.

While such a match is not entirely surprising in the deterministic $k = 0$ problem, as the one-hot encoding of the root label against which the transformer logits are compared at training corresponds to the exact marginal distribution yielded by $BP_0$, the match is highly non-trivial in the ambiguous $k > 0$ instances, where the transformer is never explicitly guided towards the correct values during training, as the one-hot encoding of the root label does not correspond to the exact marginals anymore. This calibration therefore provides a second strong piece of evidence that the transformers spontaneously implement exact inference.

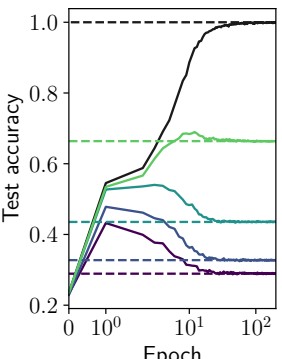

Figure 4: Evolution of the root prediction accuracy of the $k_{\text{train}} = 0$ model computed on filtered test datasets, with $k_{\text{test}} = 0, 1, 2, 3, 4$ (from top to bottom), for a model trained on $k_{\text{train}} = 0$ data and $P = 2^{17}$, $\ell = 4$, $q = 4$. The dashed lines represent the out-of-sample BP prediction.

**Supervised learning dynamics.** Looking more specifically at the learning dynamics of a network trained on the full hierarchy sheds some light on the learning process of the transformer encoder. Fig. 4 shows the evolution of the test accuracy of the $k_{\text{train}} = 0$ model both in-sample, with $k_{\text{test}} = 0$ data, and out-of-sample, on filtered data with $k_{\text{test}} > 0$. One can notice multiple stages in the learning procedure: in the first epochs, the network imputes a simplistic explanation of the training data, resolving the leaf-to-root correlations—aided by the supervised signal—, as well as the short-range correlations between the leaves. As a result, the test accuracy increases for all values of $k_{\text{test}}$. As time progresses and longer-range correlations are discovered in the training data, the accuracy on the most filtered datasets drops towards the *mismatched* $BP_0$ prediction, since the imputed higher correlation levels are not present in the out-of-sample $k_{\text{test}} > 0$ data. In the meantime, the accuracy for the smallest values of $k_{\text{test}}$ keeps increasing. In a limited number of epochs, as the network perfectly learns to infer the root on $k_{\text{test}} = 0$ data, the $BP_0$ oracle accuracy is reached on test sets generated with all levels of factorization.

This picture can be further refined by considering the predictions of a transformer trained on the full hierarchy and the evolution of their distance from the marginals predicted on the same data by the $BP_k$ oracles, for all $k \geq 0$. As illustrated by the $D_{\text{KL}}$ in Fig. 1(c), we observe an initial stronger alignment to $BP_\ell$, which only considers leaf-to-root correlations. As training on $k_{\text{train}} = 0$ data progresses and the transformer shifts towards the correct prediction, the model predictions sequentially align to versions of BP that incorporate more and more of the correlation structure—i.e., $BP_k$ with decreasing values of $k$.

### 3.3 MASKED LANGUAGE MODELING

We now turn to self-supervised training, where the model learns from a dataset of $P$ *unlabeled* sequences. In simple terms, the Masked Language Modeling (MLM) training procedure consists of randomly masking parts of the sequences and asking the model to recover them from the context. This is closer to what is done in practice to train large language models, see e.g. Devlin et al. (2019); Liu et al. (2019). While in principle one could mask several symbols simultaneously in training, we focus on single-symbol masking—at a random position in the sequence—in the following, given the limited length of our sequences (a single symbol representing already 6.25% of the sequence for $\ell = 4$). Contrary to the root inference task, in MLM perfect accuracy cannot be achieved even in the fully hierarchical case, because of the stochastic nature of the branching process in the generative tree. The optimal performance is still yielded by the BP matched to the test data.

To reconstruct the masked symbol, we now feed a single token, selected from the final transformer encoding at the positions associated with the masked element, to a linear layer producing a vector of logits. The network is then trained by minimizing the cross-entropy loss between these logits and the one-hot encoding of the masked element in the sequence.

**Optimal reconstruction performance.** Given sufficient data, we find that transformers again approach optimal in-sample accuracy on data with any level of filtering. We show the case trained on $k_{\text{train}} = 0$ in Fig. 5, where the transformer reaches the $BP_0$ accuracy also on out-of-sample test data with $k_{\text{test}} > 0$. Consistent with intuition, the required amount of training data $P^*$ is increased relative to the supervised task, as the network must learn to resolve the weak long-range correlations in the sequence without any supervised signal from the top of the hierarchy. Moreover, compared to

root classification, the networks trained for MLM require much longer training to approach optimal performance—typically $\sim 10^3$ epochs in place of a mere $\sim 10$ epochs for classification—, see Fig. 5 vs Fig. 4.

**Full prediction matching.** To go beyond test accuracy, we also consider the full probabilities outputted by the transformer. As shown in the top panel Fig. 1(b), we find a close match with the exact marginals obtained from BP when measured on in-sample inputs. To confirm the generality of this correspondence, we extend the comparison to uniformly sampled data in the bottom panel of Fig. 1(b). In this setting, we still observe high correlations between the outputs, albeit with more dispersion related to the markedly atypical nature of these test samples compared to the training data distribution. Measuring the alignment using the Kullback-Leibler divergence, shown in Fig. 1(d), or else the sample-specific prediction match and Spearman (ranking) correlation between the two discrete probability distributions, shown in Fig. 12 of Appendix D.4, confirms the near equivalence between transformer and BP computation. Note again the remarkable calibration of the logits, although the network is trained with hard labels for the masked symbols despite the probabilistic nature of the task.

**Self-supervised learning dynamics.** By analyzing the out-of-sample performance with different filtering levels, we also unveil the sequential nature of the MLM learning process. Computing the test accuracy on all $k_{\text{test}}$ levels throughout the training dynamics, we observe a clean "staircase" behavior in the test accuracy, as shown in Fig. 5. This picture confirms and clarifies the experiments in Fig. 4, showing that the network sequentially resolves the nested levels of the hierarchy, in a bottom-up order. Note that the observation of the shorter-range correlations being learned first is consistent with the signal-to-noise picture exposed in Cagnetta & Wyart (2024). Moreover, the presence of a sequential mechanism of discovery and resolution of different moments of the data distribution has been studied in Refinetti et al. (2023); Bardone & Goldt (2024); Rende et al. (2024). Overall, the convergence of the transformer to both the in-sample and the out-of-sample token prediction accuracy of BP supports the claim that the model learns to implement a close approximation of the exact algorithm. The learning mechanism is also confirmed by the behavior of $D_{\text{KL}}$ along the training, shown in Fig. 1(d): analogous to the root inference case, but more qualitatively compelling, the predictions of a transformer trained on the fully hierarchical data sequentially align with the marginals yielded by $\text{BP}_k$, with decreasing $k$ as training progresses and longer-range correlations are accounted for.

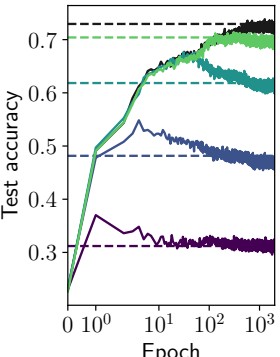

Figure 5: Evolution of the root prediction accuracy computed on filtered test datasets, with $k_{\text{test}} = 0, 1, 2, 3, 4$ (from top to bottom), for a model trained on $k_{\text{train}} = 0$ data and $P = 2^{17}$, $\ell = 4$, $q = 4$. The dashed lines represent the in- and out-of-sample $\text{BP}_0$ predictions.

## 4 HOW TRANSFORMERS EMBED THE EXACT INFERENCE COMPUTATION

**Attention map analysis.** In the root inference task, the readout performing the prediction is fed with the entire sequence of tokens. As a result, there are many ways for the transformer encoder to distribute the computation across its layers, and no necessity for single tokens to carry information on all the ancestry levels in the tree, making it a non-ideal setting for mechanistic interpretation.[2] In the MLM task, on the other hand, single token encodings are used to predict the masked symbols. This requirement seems to guide the model towards more interpretable attention maps, shedding some light on how the model may approximate the optimal algorithm. They are shown in Fig. 6, each row referring to a transformer encoder trained on data with different filtering levels—$k$ increasing from top to bottom.

In the fully filtered case (bottom row) there is no need to combine the different elements of the sequence before the readout and the attention matrices are nearly uniform. Now, as we reduce the level of filtering in the generative process, clear patterns emerge in the attention map.

---

[2]Note that transformers trained on the classification task still present some patterns related to the hierarchical nature of the data model, albeit less clearly, see Appendix D.5.

First, the model focuses on short-ranged correlations between nearest neighbors when $k = 3$ and, as we decrease $k$, towards patterns of size $\sim 2^{\ell-k}$, which is the exact size of the stronger correlated block with a filtering parameter $k$—see Sec. 2. Note that the similarity between the $k = 1$ and $k = 0$ cases (top two rows) is natural, the tree topology in these two cases being identical and with only the transition probabilities for this first layer differing.

Interestingly, the network naturally organizes the attention layers hierarchically. This is particularly visible when there are fewer redundant layers i.e. in the cases $k = 0, 1$ (two top rows in Fig. 6). Such a layout is consistent with the BP algorithm on the full tree, where one combines elements pairwise while going up the tree. While a typical BP implementation includes a downward pass, it is possible to avoid this step if the token embedding dimension, $d$, is sufficiently large. To illustrate this point, we propose an existence proof of a plausible implementation of the BP algorithm in an architecture.

**Exact transformer embedding of BP.** In a natural implementation of BP, inference for the MLM task requires the messages from the visible leaves to reach the top of the hierarchy and descend back to the masked symbol, effectively propagating through $2\ell$ layers. A proposal in Zhao et al. (2023) for a transformer embedding of the inside-outside parsing algorithm—a generalization of the above-described BP to the unknown topology setting—requires as many transformer blocks as double the sequence length—here $2^\ell$—

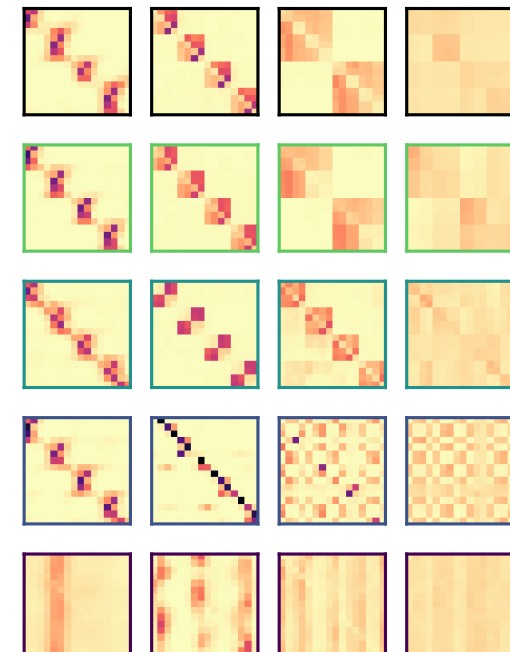

Figure 6: Visualization of the $n_L = 4$ attention matrices (averaged over $10^4$ input sequences) for transformers trained on the MLM task on different filtered datasets, with $k = 0, 1, 2, 3, 4$ (top to bottom rows), and $P = 2^{18}$, $\ell = 4$, $q = 4$. For the fully factorized model, $k = 4$, where the leaves are independent conditional to the root the attention matrix appears structureless. When $k$ decreases one sees the emergence of attention blocks of size $\leq\sim 2^{\ell-k}$. For $k = 0, 1$, the trained attention matrices reflect all the hierarchies of the correlations.

, and an attention head per hidden symbol in the hierarchy. Thus, it might seem surprising that a single-head transformer encoder with $\ell$ blocks could be sufficient to mimic the BP algorithm. To prove the feasibility of its implementation within these architectural constraints, we propose an idealized transformer implementation of the BP algorithm. Note that some of the key ingredients of this feasible implementation are introduced for the sake of interpretability but are not imposed in our experiments, and therefore this does not represent an exact explanation of the trained transformer computation. The complete existence argument is deferred to Appendix E, while here we provide a high-level description of some key ideas.

We consider a fully disentangled embedding of positional and semantic information in the vectorized tokens, contained in $d = q(q + 2) + \ell$ dimensions. The isolation of the semantic information allows the implementation of a simple position-based attention mechanism, inspired by the factor graph structure, and compatible with the attention matrices in Fig. 6. Then, going up the hierarchy requires the computation of a trace of products (see equation 4), which can be well approximated by the fully connected layers in the second part of the transformer blocks, provided the attention selects the right terms in the product. The less intuitive component of the implementation is the computation of the messages directed towards the leaves, used in the MLM task. Given the limit on the number of transformer blocks, this computation must be done in parallel with the upward climb of the hierarchy, despite the missing downward messages. It turns out that, by exploiting $\mathcal{O}(q^2)$ memory slots in the token embedding—and thus with an increased memory cost compared to BP—a different recursion with the same result as the standard message-passing can be implemented, within the $n_L = \ell$ constraint for the number of transformers layers.

**Probing the encoder representations.** To confirm that the computation going up the tree is distributed sequentially in the transformer blocks, consistent with the proposed embedding of BP, we undertake a probing experiment similar to those performed e.g. in Zhao et al. (2023). First, we analyze the encoder trained for the MLM task on $k = 0$ data, cf. top row of Fig. 6. Keeping the encoder weights *frozen*, we investigate how much information about the ancestors of any leaf is contained in the successive hidden representations of the corresponding token—see Appendix D.6 for implementation details. While in the exact embedding of BP the $k$-th level ancestor information must be available at layer $k$ to iterate the recursion for the downgoing messages, the MLM training does not set such a requirement. To probe the encodings, we employ a specialized two-layer readout for each encoder-layer/ancestry-level pair—independent of the token position—trained on a supervised dataset with $2^{14}$ examples. In Fig. 7, we show that the prediction accuracy is high on ancestors up to the same level as the probed layer and deteriorates on higher levels of ancestry. Note that, unless the information about the entire block of $2^{\ell-k}$ tokens is properly mixed in through the attention mechanism, a perfectly accurate prediction of the common $k^{th}$ level ancestor from a single token representation is impossible, as the mapping becomes non-deterministic. Moreover, the "overfitting" sce-

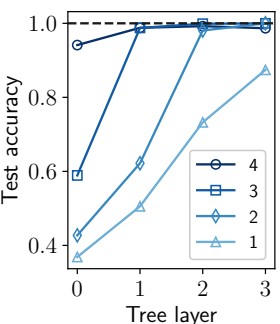

Figure 7: Test accuracy in the ancestor prediction task (layer 0 is the root) with $\ell = 4$, $q = 4$, $k = 0$ obtained by reading out the intermediate transformer encoding levels (legend) of a model pre-trained on the full hierarchy. The readout is trained on $2^{14}$ labeled examples.

nario, where the ancestors are reconstructed solely by the trained probes and the sequential reconstruction is an artifact, can be ruled out by considering the gap between the accuracies achieved from different layers—the relative comparisons are fair since the readouts are trained on the same datasets—, and by training the probes only on some positions—see Appendix D.7.

In Appendix D.7, we also conduct similar ancestor prediction experiments on the last encoder layer of models trained with $k > 0$ data (lower rows of Fig. 6), where we again find that the ancestry information is consistent with the attention maps.

**Synergy between tasks and MLM pre-training.** In the context of our model, we can straightforwardly explain why self-supervised pre-training allows a large speed-up in the supervised training process, in line with many empirical observations on real-world data (Howard & Ruder, 2018). We show in Fig. 1(f) an MLM pre-trained model fine-teuned for root inference. A significant reduction in the labeled data required to achieve optimal root inference —$P^*$ in Sec. 3.2— is observed, both with frozen and with fine-tuned encoder weights.

## 5 CONCLUSIONS

By using a simple, tunable, hierarchical model of structured sequences, we were able to shed some light on the inner workings of transformer encoders and better understand how they achieve optimal inference on both supervised and self-supervised tasks. The modularity of our data model also allowed us to uncover how transformers sequentially implement longer-range correlations during the learning dynamics, compatible with similar controlled studies (Rende et al., 2024) and with the general understanding of LLMs trained on natural language (Kaplan et al., 2020). This mechanism could perhaps be exploited to shape theory-driven curriculum learning strategies for NLP, where curating the presentation order of training examples was already proven effective (Campos, 2021).

Generalizing our filtering-based interpretative tool to the case of variable sequence lengths (Allen-Zhu & Li, 2023; Zhao et al., 2023)—where the topology of the parsing tree is not known *a priori*—is a challenging but promising direction for approaching a more detailed understanding of the learning dynamics and the embedded computation in transformer trained on natural language. On the other hand, while the idealized model of structured sequences studied in the present work might be less suited for modeling natural language compared to standard CFGs, the agnostic nature of the approach could open connections to other related fields, like protein sequences analysis (Zhang et al., 2023) and immunology (Meynard-Piganeau et al., 2024). It could finally be interesting to undertake a similar investigation on the way transformers learn in other problems where optimal inference can also be achieved via dynamic programming (Mossel et al., 2014; 2023).

REPRODUCIBILITY STATEMENT

We provide the source code used to perform our numerical experiments in the Supplementary Material (SM). It includes a Python script generating the data, as well as the PyTorch implementation of the transformer and training scripts for both root inference and MLM. It finally provides an efficient implementation of the Belief Propagation algorithm which can be used for both root inference and Masked Language Modeling. The data used to produce the figures in the main text corresponds to fixing `seed = 0` and `sigma = 1` in the data generation script, see Appendix A for details on the role of the latter.

## A FURTHER DETAILS ON OUR DATA MODEL

The transition tensor **M**—the "grammar" of our generative model in CFG terminology—fully controls the properties of the above-defined generative process. We define a parametrized ensemble of random grammars, from which multiple transition tensors can be sampled independently. Two grammars generated with the same parameters are expected to share some high-level features and produce data of comparable complexity, at least in the large vocabulary size limit. Elaborating on recent work on context-free grammars (see Sec. 2.3 of the main text), we generate transition probabilities as

$$M_{abc} = \frac{e^{h_{abc}}}{\sum_{b'c'} e^{h_{ab'c'}}} \tag{7}$$

where the logits $h_{abc}$ are generated as

$$h_{abc} = \begin{cases} \sigma \xi_{abc} & \text{if} \quad (b,c) \in \mathcal{O}_a, \\ -\infty & \text{otherwise,} \end{cases} \tag{8}$$

with $\xi_{abc}$ independent Gaussian random variables of zero mean and unit variance, and $\sigma$ controlling the probability fluctuations between likely and unlikely transitions. Here, the $q$ sets $\mathcal{O}_a$ build a equal-sized partition of the $q^2$ possible children pairs $(b,c)$, i.e. $\mathcal{O}_a \cap \mathcal{O}_{a'} = \emptyset$ if $a \neq a'$ and $|\cup_a \mathcal{O}_a| = q^2$. This non-overlapping prescription implies that the broadcast from the root to the leaves has no ambiguity. Therefore, as stated in the main text, if the transition tensor **M** is known, one can deterministically go up the hierarchy of the tree and infer the root given a set of leaves. We leave generalizations of this setting for future work.

## B VANILLA ENCODER-ONLY TRANSFORMER ARCHITECTURE

A sequence of leaves $\{x_i\}$ generated by the hierarchical model and represented by $2^\ell$ integers is first converted into a sequence of one-hot vectors $\{\boldsymbol{x}_i\}$, with $\boldsymbol{x}_i \in \mathbb{B}^q$. [3] Then, we perform the first encoding step producing a sequence of *tokens* $\{\boldsymbol{x}_i^{(0)}\} \in \mathbb{R}^d$, with arbitrary dimension $d \geq q$, obtained through a learnable projection to the embedding space and the inclusion of positional encoding $\boldsymbol{p}_i$,

$$\boldsymbol{x}_i^{(0)} = \boldsymbol{W}_E \boldsymbol{x}_i + \boldsymbol{p}_i, \tag{9}$$

with $\boldsymbol{W}_E \in \mathbb{R}^{d \times q}$ and $\boldsymbol{p}_i \in \mathbb{R}^d$. For our experiments, we consider $d = 128$ and the standard sinusoidal positional encoding of Vaswani et al. (2017).

As described in the main text, each transformer block in the network then transforms the tokens as follows,

$$\tilde{\boldsymbol{x}}_i^{(l)} = \text{layernorm}\left(\boldsymbol{x}_i^{(l-1)} + \text{selfattention}(\boldsymbol{x}^{(l-1)}; \boldsymbol{W}_Q^{(l)}, \boldsymbol{W}_K^{(l)}, \boldsymbol{W}_V^{(l)})\right), \tag{10}$$

$$\boldsymbol{x}_i^{(l)} = \text{layernorm}\left(\tilde{\boldsymbol{x}}_i^{(l)} + \text{FC}(\tilde{\boldsymbol{x}}_i^{(l)}; \boldsymbol{W}_1^{(l)}, \boldsymbol{W}_2^{(l)})\right). \tag{11}$$

The single-head self-attention layer considered in this work entails the computation of three different quantities from each token: the query $\boldsymbol{q}_i = \boldsymbol{W}_Q \boldsymbol{x}_i$, the key $\boldsymbol{k}_i = \boldsymbol{W}_K \boldsymbol{x}_i$ and the value $\boldsymbol{v}_i = \boldsymbol{W}_V \boldsymbol{x}_i$.

---

[3]For simplicity, the procedure described here does not consider special tokens. In practice, we will take a vocabulary of size $q + 1$ to account for *masked symbols* when doing MLM, see Devlin et al. (2019).

For simplicity, we take $\boldsymbol{W}_Q$, $\boldsymbol{W}_K$ and $\boldsymbol{W}_V$ in $\mathbb{R}^{d \times d}$. The queries and keys are combined to compute the attention matrix

$$A_{ij} = \text{softmax}\left(\frac{\boldsymbol{q}_i \cdot \boldsymbol{k}_j}{\sqrt{d}}\right), \tag{12}$$

then used to build a linear combination of the values,

$$\text{selfattention}(\boldsymbol{x}; \boldsymbol{W}_Q, \boldsymbol{W}_K, \boldsymbol{W}_V) = \sum_{j=1}^{2^\ell} A_{ij} \boldsymbol{v}_j. \tag{13}$$

The fully-connected layer, instead, is a standard 2-layer network with relu activations:

$$\text{FC}(\boldsymbol{x}_i; \boldsymbol{W}_1, \boldsymbol{W}_2) = \boldsymbol{W}_2 \,\text{relu}\left(\boldsymbol{W}_1 \boldsymbol{x}_i\right), \tag{14}$$

where $\boldsymbol{W}_1 \in \mathbb{R}^{d \times d'}$, $\boldsymbol{W}_2 \in \mathbb{R}^{d' \times d}$, and $d' = 2048$ in our experiments. We refer the reader to the original paper by Vaswani et al. (2017) for additional details on the transformer encoder operations.

## C   FURTHER DETAILS ON NUMERICAL EXPERIMENTS

All numerical experiments presented in this paper were performed using PyTorch (Paszke et al., 2019) version 2.3.0. We use the Adam (Kingma & Ba, 2014) optimizer with batches of size 32 and a fixed learning rate of $10^{-4}$, other parameters left as default. We did not find learning rate scheduling to provide significant benefits in our experiments. All models were initialized randomly using the default settings (Xavier uniform distribution).

In both root inference and MLM, the accuracy of the transformer implementation and of the BP over $M$ trials is measured straightforwardly as

$$\text{Accuracy} = \frac{1}{M} \sum_{\gamma=1}^{M} \delta_{\hat{x}_\nu, x_\nu}, \tag{15}$$

where $x_\nu$ is understood as the ground truth and $\hat{x}_\nu$ the symbol inferred using the network or BP.

The Kullback-Leibler divergence between two discrete probability distributions encoded as $n$-dimensional vectors $\boldsymbol{u}$ and $\boldsymbol{v}$, is given by

$$D_{\text{KL}}(\boldsymbol{u} \parallel \boldsymbol{v}) = \sum_{\alpha=1}^{n} u_\alpha \, \log\left(\frac{u_\alpha}{v_\alpha}\right). \tag{16}$$

## D   ADDITIONAL FIGURES

### D.1   INFLUENCE OF THE NUMBER OF ATTENTION LAYERS

Establishing a relation between the number of encoder layers $n_L$ in the transformer and the ability to achieve this optimal classification on data generated from hierarchical models is also not straightforward. Indeed, given the concatenation of operations involved in a single transformer block and the presence of residual and normalization layers, the effective number of computational layers in a transformer is not as explicit as in a multilayer perceptron or a CNN architecture. As apparent in the main text, setting $n_L = \ell$—or $n_L \geq \ell - k$ for filtered data—enables the transformer to converge towards a very interpretable parameter configuration. However, this natural choice does not appear to be strictly necessary for the transformers to achieve optimal inference, at least when the number of embedding dimensions $d$ is large.

More specifically, Fig. 8 shows that the test accuracy on the root classification task on $k = 0$ unfiltered data can reach the optimal value for $n_L < \ell$. While $n_L = \ell = 4$ is the most sample efficient, it is clear that $n_L = 3$ provides comparable performance, and only $n_L = 1$ appears to lead to poor sample efficiency. In all the performed experiments, a bigger value for $n_L$ corresponded to better sample efficiency, which seems to indicate that more flexible models require less data to reach the same performance level despite the increased number of parameters to train.

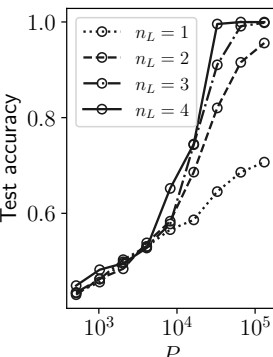

Figure 8: Reproduction of Fig. 1(b) with now $n_L \leq 4$ attention layers in the transformer encoder and restricted to the "worst case" $k = 0$ unfiltered dataset.

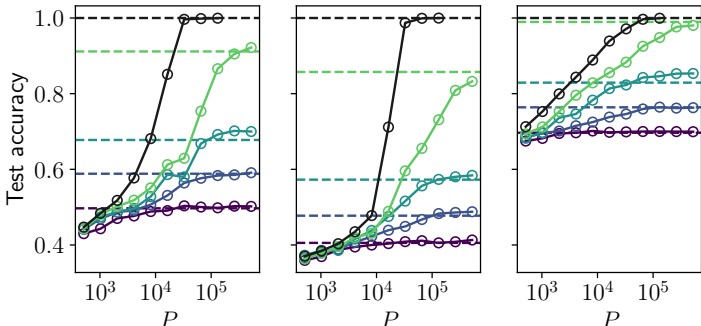

Figure 9: Reproduction of Fig. 1(b) on other realizations of the transition tensor **M** for the same parameters $\ell = 4$, $q = 4$, $\sigma = 1$. We remind that for the $k > 0$ cases, the BP predictions (dashed lines) are not Bayes optimal, as the test accuracy is measured out-of-sample here. From left to right, these grammars can be reproduced by fixing `seed = {1,15,31}` in the data generation code provided in the SM.

In any case, the required complexity of the architecture is clearly related to the amount of structure in the data model. As an extreme illustration, in the case of fully filtered correlations $k = \ell$, the BP marginals for the root are just products of conditional probabilities on the leaves as $P(x_0 = a \mid \{x_i\}) \propto \prod_{i=1}^{2^\ell} P(x_i \mid x_0 = a)$, i.e. a "Naive Bayes" classifier is optimal. Any layer of attention is thus superfluous since a standard feed-forward network with a single hidden layer is sufficient for this task. In fact, the analysis of the attention maps (trained this time on MLM) in Sec. 4 confirms this natural intuition, as most attention layers appear effectively unused by the transformer when $n_L > k$.

## D.2 OTHER GRAMMARS

As expected from the log-normal nature of its entries, there may be significant sample to sample fluctuations in the transition tensor **M** for a given value of $\sigma$, which we expect to (slowly) decay as $q$ becomes large. All the results presented in the main text come from the same grammar with $q = 4$, $\sigma = 1$ (corresponding to `seed = 0` in the data generation script provided in the SM, see the Reproducibility Statement above), however we illustrate that all our conclusions should qualitatively hold for any realizations of **M** in Fig. 9. Indeed, while there are some very clear differences in the "difficulty" of the grammars presented, the transformer architecture performs very similarly, here on the root inference task. All subsequent experiments can be reproduced on these different grammars, yielding an unchanged phenomenology.

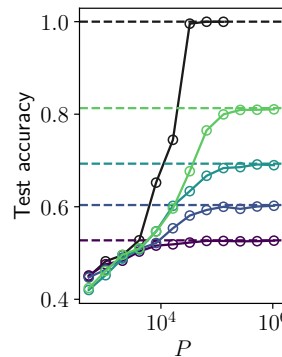

Figure 10: Reproduction of Fig. 3 with the test accuracy computed on (in-sample) factorized data, rather than the out-of-sample testing presented in the main text.

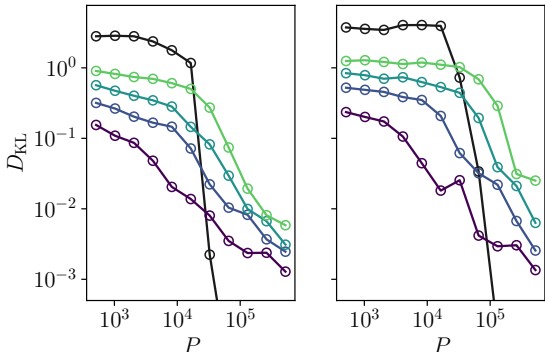

Figure 11: Reproduction of Fig. 10 with the Kullback-Leibler divergence between the transformer outputs BP marginals for identical levels of factorizations for (Left) in-sample inputs, (Right) uniformly randomly generated inputs.

### D.3 IN-SAMPLE CLASSIFICATION PERFORMANCE ON FILTERED DATASETS

Fig. 10 shows the test accuracy computed in-sample for the factorized datasets as a function of the training set size $P$. The optimal inference accuracy predicted by the Belief Propagation, which is not unity when $k > 0$, is reached by the transformers in all cases when trained on sufficient data.

It appears that the required amount of data $P^*$ for reaching optimal accuracy not only depends on the specific transition tensor $\mathsf{M}$ (see Fig. 9 for an illustration for $k = 0$), but also on the level of factorization. For intermediate values of $k$, $P^*$ is notably larger than with the $k = 0$ full hierarchy. This is due to the fact that the $k = 0$ case is quite unique for two (related) reasons. The first is that the logits outputted from the network need not be calibrated, so the accuracy can reach the optimum without the transformer having fully implemented an algorithm equivalent to BP, whereas the relative weights of prediction must be well understood to match the optimal inference in the ambiguous $k > 0$ cases—in other words it is easier to match perfect accuracy with approximate weights when the true distribution is $\delta$-distributed. The other is that this being said, matching the BP is also easier in the $k = 0$ case because it is the only case where the training cross-entropy loss corresponds exactly to that computed with the true marginals—that are also delta distributed due to the determinism of the task—whereas in the $k > 0$ cases the training loss does not guide explicitly to the exact marginals. The latter clearly appears in Fig. 11, showing the Kullback-Leibler divergence between the transformer outputted logits and the BP marginals instead of the test accuracy.

Note that the other case which has a singularly small sample complexity is that of the fully filtered data, $k = \ell$, as it is implementable in a single feedforward layer and does not require an implementation equivalent to BP.

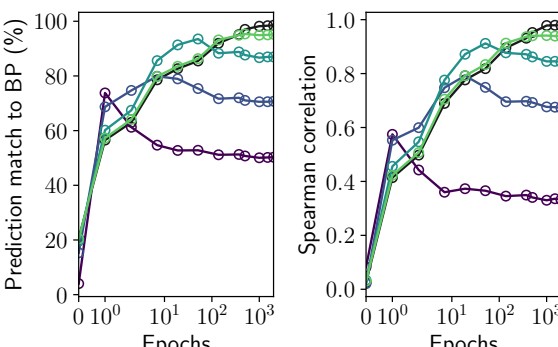

Figure 12: Reproduction of Fig. 1(d) with the prediction (i.e. $\arg\max$) match (left) and Spearman (i.e. ranking) correlation (right) between the transformer outputs and BP marginals.

## D.4 ADDITIONAL COMPARISON OF THE OUTPUTS

For completeness, we show the comparison between the full transformer predictions and the BP marginals through MLM training using the percentage of matches in the largest value (i.e. prediction match) and the spearman (ordering) correlation in Fig. 12. These confirm the observations described in the main text.

## D.5 CLASSIFIER ATTENTION MAPS

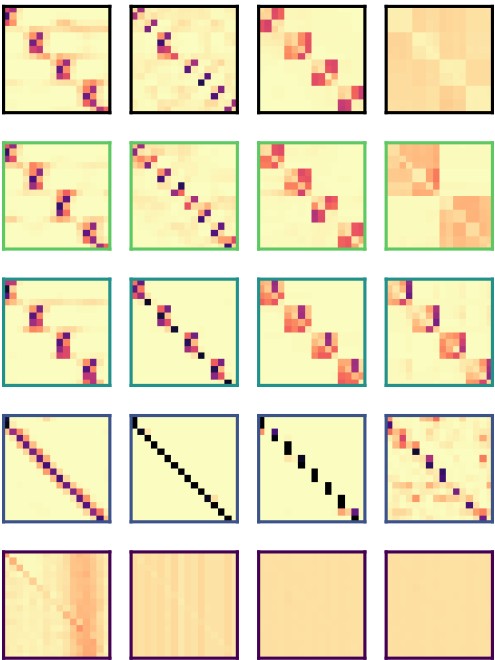

Figure 13: Reproduction of Fig. 6 for the supervised task on filtered datasets of size $P = 2^{17}$ for $k = 0$ and $P = 2^{20}$ for $k > 0$.

Fig. 13 shows the attention maps resulting from the supervised training for transformers achieving the optimal performance on datasets with different filtration levels. As in the masked language modeling task, one immediately notices the emergence of blocks of size $\sim 2^{\ell - k}$. In this prescription, where tokens are not required to be fully descriptive, it is however difficult to identify a clear pattern relating to the distribution of the computation across the different layers.

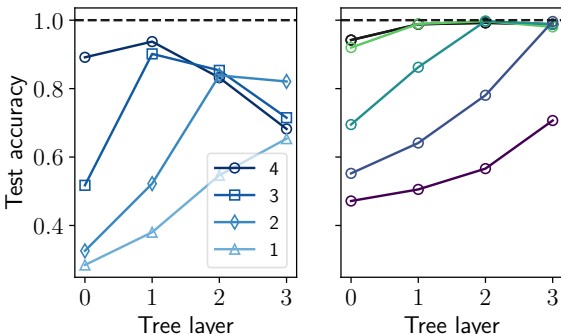

Figure 14: (Left) Reproduction of the probing experiment presented in Fig. 7, with the readout trained only on the first and last token embeddings of the sequences and tested on all elements. (Right) Test accuracy in the ancestor prediction task (layer 0 is the root) with $\ell = 4$, $q = 4$, $k_{\text{test}} = 0$, obtained by reading out the complete transformer encoding of models pre-trained $k_{\text{train}} = 0, 1, 2, 3, 4$ (from top to bottom), i.e. using the attention maps illustrated in Fig. 6 The readout is trained on $2^{14}$ labeled examples.

### D.6 DETAILS ON THE PROBING EXPERIMENTS

In order to perform the experiments presented in Fig. 7, we replace the linear readout of a trained MLM transformer by a two-layer feedforward network with 64 hidden units, acting *independently* on all of the $d$-dimensional sequences ($d = 128$ in all of our experiments, see Sec. 3) outputted by the *frozen* transformer encoder. The training of the readout is performed on $2^{14}$ labeled sequences, the labels being, for each of the elements of the sequence, the symbol on the relevant ancestor in the generative tree. Here again, the loss is taken to be the cross-entropy between the logits outputted by the network for each token and their correct ancestor label, then averaged on all the sequence elements. We present another experiment, where the cross-entropy is measured only with the first and the last token embeddings of the sequence, just below. The readout is trained on 100 epochs in all cases, which we found to be sufficient for the relatively small training set size we used.

### D.7 FURTHER PROBING EXPERIMENTS

To complement and contextualize the probing experiments presented in the main text, we provide two additional experiments. In the left panel of Fig. 14, we perform the same experiment as in Fig.7, but with probes trained only on two positions in the token sequence (first and last) and tested across all positions. While some accuracy is lost, since the readout cannot fully disentangle the positional information from the semantic one in positions that were never seen at training, the sequential effect is still evident. Moreover, we also performed the same procedure as Fig. 7 on the tokens' hidden representations, but with models trained on factorized data. As visible in the right panel of Fig. 14, a model trained of factorized data can only accurately recover ancestors up to the level in which factorization kicks in. For example, in an $l = 4$ tree, a model trained on $k_{\text{train}} = 2$ data can only predict ancestors up to level 2 (two ancestry layers above the leaves - above that, the tree is factorized), while a model trained on $k_{\text{train}} = 3$ can only predict ancestors up to level 3 (the ancestors right above the leaves - for the same reason). This is exactly what could be expected from the attention maps of Fig. 6. As before, we are probing the hidden representations of individual tokens, so this happens because the attention must provide mixing between $\sim 2^{\ell-k}$ elements of the sequence in order for individual tokens to carry information up to the level $k$ of the fully hierarchical generative model.

## E A POSSIBLE TRANSFORMER IMPLEMENTATION OF BELIEF PROPAGATION

We show here how the BP algorithm for leaf inference can be implemented using $\ell$ layers of transformers with token sizes which are compatible with what is used in our experiments. We consider the "worst case" scenario of a complete, unfiltered tree generative process of depth $\ell$.

**Token embedding.** We propose an implementation that relies on vectorized tokens with a structure of the form

$$
\boldsymbol{x}_i^{(m)} = \begin{bmatrix} \boldsymbol{r}_i^{(1,m)} \\ \vdots \\ \boldsymbol{r}_i^{(q,m)} \\ \boldsymbol{m}_i^{(m)} \\ \overline{\boldsymbol{m}}_i^{(m)} \\ \tilde{\boldsymbol{p}}_i \end{bmatrix}, \tag{17}
$$

where:

- $i \in \{1, ..., 2^\ell\}$ is the index of a leaf
- $m \in \{1, ..., \ell\}$ is the index of a transformer layer
- $\boldsymbol{r}_i^{(1,m)}, \ldots, \boldsymbol{r}_i^{(q,m)}$ are $q$ vectors of dimension $q$ ($q^2$ elements in total) storing the quantities needed to compute the final leaf marginals,
- $\boldsymbol{m}_i^{(m)}$ is a vector of size $q$ storing the up-going message for the ancestor of leaf $i$ at level $m$,
- $\overline{\boldsymbol{m}}_i^{(m)}$ is a vector of size $q$ storing the up-going message for the $m^{\text{th}}$ complementary ancestor of leaf $i$, see Fig. 15,
- $\tilde{\boldsymbol{p}}_i$ is a $\ell$-dimensional binary vector containing positional information on the full path from root to leaf $i$ (see below).

In this prescription, the total dimension of each token is therefore $d = q^2 + 2q + \ell$.

**Initialization.** We are going to consider the following initialization,

$$
\left( \boldsymbol{r}_i^{(a,0)} \right)_b = \frac{1}{q}, \quad \forall a, b = 1, \ldots, q, \tag{18}
$$

$$
\overline{\boldsymbol{m}}_i^{(0)} = \boldsymbol{0}, \tag{19}
$$

while the messages $\boldsymbol{m}_i^{(0)}$ should be initialized as in the standard BP given a sequence, i.e. with a Kronecker $\delta$ for known symbols and a uniform vector for masked leaves. The positional vector $\tilde{\boldsymbol{p}}_i$ should finally be a binary $\pm 1$ vector representing the sequence of left/right turns from the root to leaf $i$ (as $\sigma$ in equation 1).

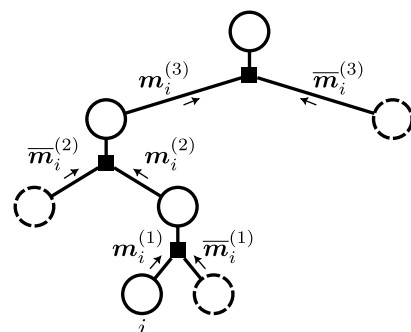

Figure 15: Illustration of the upgoing messages embedded in the tokens of the transformer implementation of BP for a tree with $\ell = 3$. Complementary ancestors are shown with dashed lines.

**Attention layer.** In our implementation, the dot product

$$
\left( \boldsymbol{W}_Q^{(m)} \boldsymbol{x}_i^{(m)} \right)^\top \left( \boldsymbol{W}_K^{(m)} \boldsymbol{x}_j^{(m)} \right)
$$

entering the softmax and at the heart of the attention mechanism only encodes positional information; more precisely, it combines the common ancestors of tokens $i$ and $j$ down to layer $\ell - m$ of the generative tree. This can be achieved with query and key matrices such that $\left( \boldsymbol{W}_Q^{(m)} \right)^\top \boldsymbol{W}_K^{(m)}$ has elements equal to zero except in its lower right corner of size $\ell \times \ell$ which has the following structure:

$$
\begin{bmatrix} \beta \mathbf{1}_{(\ell-m-1)\times(\ell-m-1)} & 0 & \boldsymbol{0} \\ 0 & -\beta & \boldsymbol{0} \\ \boldsymbol{0} & \boldsymbol{0} & [\boldsymbol{0}]_{m \times m} \end{bmatrix}, \tag{20}
$$

with $\beta \gg 1$. Let us detail the role of this $\ell \times \ell$ sub-matrix. Its upper left terms proportional to $\beta$ will be relevant in the softmax, when $\beta \gg 1$, if they are positive, meaning these are common ancestors to tokens $i$ and $j$, and negligible if they are negative. The diagonal term proportional to $-\beta$ requires

the two considered tokens to be in different positions in the sequence to contribute to the softmax, ensuring there is no influence of the messages on themselves in the following steps. Its lower right corner, which is populated by a $m \times m$ matrix of zeros, ensures that layers below $\ell - m$ in the generative tree are no longer considered.

On the other hand, the value matrix may be used to select the correct messages in the token vector, with zeros elsewhere.

As a result, the total operation amounts to averaging the message incoming from the complementary sub-tree over all the trajectories within the complementary sub-tree

$$\text{selfattention}(\boldsymbol{x}^{(m)}; \boldsymbol{W}_Q^{(m)}, \boldsymbol{W}_K^{(m)}, \boldsymbol{W}_V^{(m)})_i \approx \begin{bmatrix} 0 \\ \vdots \\ \mathbb{E}_{j \in \overline{\mathcal{S}}_i^{(m)}} \left[ \boldsymbol{m}_j^{(m)} \right] \\ \vdots \\ 0 \end{bmatrix} = \begin{bmatrix} 0 \\ \vdots \\ \overline{\boldsymbol{m}}_i^{(m)} \\ \vdots \\ 0 \end{bmatrix}, \quad (21)$$

where $\overline{\mathcal{S}}_i^{(m)}$ is the set of tokens belonging to the complementary tree of token $i$ at layer $\ell - m$ of the generative tree. Note that in principle it is not necessary to average since all of the paths should lead to the same message from the complementary tree, however keep in mind that in practice some tokens will be masked. The averaging procedure therefore allows recovering the information (unless *all* of the tokens in $\overline{\mathcal{S}}_i^{(l)}$ happen to be masked). Thanks to the skip connections, this contribution is added to the initial token, populating the initially empty entries of these complementary messages while leaving the rest of the tokens unaffected.

**Fully connected feedforward layer.** Following the initialization and after the attention layer, the encoded token has the correct structure of equation 17. One must now update the relevant information in order to go to the next attention layer and therefore the next layer in the generative tree. More precisely, we need to:

- Compute the messages of the $m + 1^{\text{th}}$ ancestor,

- Update the quantities needed to compute the marginal for the leaf associated with the token considered,

- Remove temporary or unwanted quantities stemming from the previous steps.

All of these must be done with an identical operation for all tokens as the feedforward layer is applied independently for all positions in the sequence.

The first part is to update the messages following the equivalent of equation 4,

$$\left( \boldsymbol{m}_i^{(m+1)} \right)_a \propto \sum_{bc} M_{a\mathcal{P}_i(b,c)} \left( \boldsymbol{m}_i^{(m)} \right)_b \left( \overline{\boldsymbol{m}}_i^{(m)} \right)_c, \quad (22)$$

where $\mathcal{P}_i(b, c)$ is either $bc$ or $cb$ depending on the topology of the factor node at which the update takes place—a piece of information fully contained in $\tilde{\boldsymbol{p}}_i$. This type of operation should be implementable, at least approximately, by a two-layer network since it is known to be a universal approximator.

Now, we are to compute the actual leaf marginals. As mentioned in the presentation of the standard BP implementation (Sec. 2.4), the standard approach is to perform both an upwards and downwards pass, which would require $2\ell$ attention layers.

Here, we instead wish to perform the computation in $\ell$ step, as we have seen from experiments that the transformer can achieve perfect accuracy with $\ell$ attention layers and that it does not appear to use all layers when $k < \ell$. To do so, we have included the $q^2$ elements of $\boldsymbol{r}_1^{(l)}, \ldots, \boldsymbol{r}_q^{(l)}$ in the token and now show how to update these. Note that if we had $2\ell$ layers, we could instead only store $q$ quantities.

As an example, consider the factor graph in Fig. 15 and assume the root is not pinned. We can start from the standard BP recursion for the down-going message received by leaf $i$:

$$\left(\hat{\boldsymbol{m}}_i^{(1)}\right)_{b_1} \propto \sum_{a_2,c_1} \left(\sum_{a_3,b_2} \left(\sum_{a_4,c_3} \left(\overline{\boldsymbol{m}}_i^{(3)}\right)_{c_3} M_{a_4 a_3 c_3}\right) \left(\overline{\boldsymbol{m}}_i^{(2)}\right)_{b_2} M_{a_3 b_2 a_2}\right) \left(\overline{\boldsymbol{m}}_i^{(1)}\right)_{c_1} M_{a_2 b_1 c_1}$$

(23)

and define an auxiliary message with a double index dependence:

$$\left(\boldsymbol{r}^{(a_2,1)}\right)_{b_1} = \sum_{c_1} \left(\overline{\boldsymbol{m}}_i^{(1)}\right)_{c_1} M_{a_2 b_1 c_1}.$$

(24)

In particular, the idea is that we are tracing only over the index of the complement ancestor—which is already available from the first layer—but not on the index of the downgoing message, which can only be computed after reaching the top of the hierarchy. Instead, we keep in memory all the separate contributions for each parent index. Then, we can obtain a recursion for the auxiliary messages:

$$\left(\boldsymbol{r}_i^{(a,m+1)}\right)_b \propto \sum_{h,k} M_{b\mathcal{P}_i(h,k)} \left(\boldsymbol{r}_i^{(a,m)}\right)_h \left(\overline{\boldsymbol{m}}_i^{(m)}\right)_k,$$

(25)

with the base case given in Eq. 24 treated in the transformer first layer. At the last transformer layer, one can also trace over the root index, completing the recursion. Doing so in the final feedforward layer notably yields, at the end of the transformer encoder,

$$\sum_b \left(\boldsymbol{r}_i^{(a,\ell)}\right)_b \propto \sum_{h,k} \left(\sum_b M_{b\mathcal{P}_i(h,k)}\right) \left(\boldsymbol{r}_i^{(a,\ell-1)}\right)_h \left(\overline{\boldsymbol{m}}_i^{(\ell-1)}\right)_k,$$

(26)

which is proportional to the incoming message on the leaf and therefore to its marginal if it is to be inferred. The final linear readout may then select this relevant part of the outputted tokens to perform the masked language modelling.

**Including intermediate layers.** In principle, one could add $q \times (\ell - 1)$ new vectors entries in the token in order to store the marginals at intermediate layers. These would simply be used to store the intermediate values of the $\sum_b \left(\boldsymbol{r}_i^{(a,l)}\right)_b$.

**Accommodating for filtration.** The implementation described above considered the case of $k = 0$, unfiltered generative trees, i.e. the most complex case from the BP standpoint. In the case of a dataset with filtering parameter $k$, one can adapt the implementation by taking $\ell - k$ layers. The central difference then lies in the $\ell - k^{\text{th}}$ block, which must then combine the $2^k$ messages going up to the root in its feedforward layer (instead of two messages like at all other layers in the $k = 0$ case).

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
