# OpenReview forum: "How transformers learn structured data: insights from hierarchical filtering"
_ICLR.cc/2025/Conference — Submitted to ICLR 2025_

### Official Review · Reviewer_BYbw · 2024-10-23

**Soundness:** 3
**Presentation:** 2
**Contribution:** 2
**Rating:** 5
**Confidence:** 2

**Summary:**

The authors build a binary-tree CFG with a filtering mechanism that the nodes on layer k is sampled only conditioned on the root node.

A transformer encoder is then trained to predict the root node. With k=0, the prediction is perfect. And the performance decreases as k increase. It is also shown that the model's performance on out-sample k exactly matches the performance from BP. The authors then conduct experiments on masked prediction (like BERT). Finally, the authors propose an exact implementation of the BP algorithm through the transformer computation.

**Strengths:**

It's a novel viewpoint to study transformer learning from belief propagation.

The CFG construction with filtering is interesting.

The study is from multiple perspective: prediction accuracy, probing, and manual construction.

**Weaknesses:**

I'm not sure whether the transformer accuracy matches that of belief propagation is surprising, could it be a natural consequence that the transformer is simply learning the "optimal" prediction function (which is the prediction made by belief propagation)?

I think the writing of this paper can be improved.

While the construction sec3.5 is interesting, it does not mean that the learned transformer is actually doing that (if I understand correct).

I may consider raise my score if other two reviewers show strong interest.

**Questions:**

Line355 I did not quite understand why the accuracy would have a drop in the middle of training.

Also I hope the writing for section3.3 can be improved by giving more easy-to-understand intuition.

---

> ### Author Response · Authors · 2024-11-19
> **Author response (part 1)**
>
> We thank the reviewer for carefully reading our work and providing valuable feedback, which we will use to improve our presentation. We would first like to address the overall weaknesses identified by the reviewer:
>
> **(W1) I'm not sure whether the transformer accuracy matches that of belief propagation is surprising … transformer is simply learning the "optimal" prediction function**. We thank the referee for his comment, and we agree, in a sense. As stated in our introduction, transformers assimilate natural language that is vastly more complex than our data model. What we think is key in our study is understanding how it does so, both in time during learning, capturing the longer range correlations through a clear staircase identified thanks to the filtering, as well as in ‘space’ within the architecture, organizing the computation throughout its layers in an interpretable way, strikingly similar to the most obvious implementation of BP in l layers. In the context of this data model, BP is the exact (and thus optimal) oracle, so we will rephrase “learning to implement BP” as “learning to perform optimal inference”.
>
> **(W2) I think the writing of this paper can be improved.** We are currently working on streamlining the first half of our paper, in particular, to be more straight to the point. We will clearly highlight our contributions, which have been received with some confusion. Moreover, we added some additional experiments, that strengthen our claims in regards to the model approaching the implementation of the exact inference algorithm.
>
> **(W3) While the construction sec3.5 is interesting, it does not mean that the learned transformer is actually doing that.** This is entirely true, and we have added a line highlighting that this is an existence proof for the implementability of BP in an l-layer transformer, but we are not claiming that it is exactly being implemented. In fact, the described implementation is meant to be understandable and requires, for example, full disentanglement between positional and semantic information, which is not forced in our experiments. Nonetheless, we believe the existence of this implementation is non-trivial, and that it strengthens our paper significantly relative to other works that have attempted to study the parsing of CFGs in transformers, since it can be used in practice as a tool for interpretability. The state-of-the-art work of Zhao et al. (2023), notably only provides a possible implementation that requires more attention layers than BERT has. We leave a precise study of the exact relation between the theoretical implementation vs the learned implementation for future work, but it is easy to notice that the organization of the attention layers qualitatively agrees between the two, suggesting a strong link.

---

> ### Author Response · Authors · 2024-11-19
> **Author response (part 2)**
>
> To address the referee’s questions:
>
> **(Q1) Why the accuracy would have a drop in the middle of training.** This behavior is very important as part of our understanding of the learning dynamics, so we regret that we have not successfully communicated this point (as pointed out also by reviewer VwvV). We are working on clarifying it. In a nutshell, the accuracy goes up and down on test sets generated from different (higher) filtering data than the training set, and therefore OOD from the perspective of the training model. The explanation comes from the fact that the transformer discovers the existence of higher hierarchical correlation levels (i.e., longer-range correlations) sequentially, during training. It starts by imputing a simplistic explanation for the data (e.g., a correlation length of 2), thus initially increasing its accuracy on the corresponding OOD data. But then an additional correlation level (e.g., correlation length of 4) is imputed, and the transformer stirs its predictions accordingly. At this point, the accuracy on the simpler, more factorized data decreases, since the model is learning to assume a richer correlation structure between the symbols, which is not present in the simpler OOD data. This staircase behavior is novel in our work and may be a more general feature of deep network learning dynamics (Refinetti et al., 2023; Bardone & Goldt, 2024; Rende et al., 2024).
>
> **(Q2) The writing for section 3.3 can be improved by giving more easy-to-understand intuition.** We are making an effort to improve our presentation, putting stronger accents on the main results and on their discussion, and simplifying the technical details sections.
>
> **References:**
>
> Bardone, L., & Goldt, S. (2024). Sliding down the stairs: how correlated latent variables accelerate learning with neural networks. arXiv preprint arXiv:2404.08602.
>
> Refinetti, M., Ingrosso, A., & Goldt, S. (2023). Neural networks trained with SGD learn distributions of increasing complexity. In International Conference on Machine Learning (pp. 28843-28863). PMLR.
>
> Rende, R., Gerace, F., Laio, A., & Goldt, S. (2024). A distributional simplicity bias in the learning dynamics of transformers. arXiv preprint arXiv:2410.19637.
>
> Zhao, H., Panigrahi, A., Ge, R., & Arora, S. (2023). Do transformers parse while predicting the masked word?. arXiv preprint arXiv:2303.08117.

---

> > ### Comment · Reviewer_BYbw · 2024-11-22
> > **Thank you**
> >
> > Thank you for the response, I'll maintain my current rating for now.

---

### Official Review · Reviewer_VwvV · 2024-10-30

**Soundness:** 2
**Presentation:** 3
**Contribution:** 2
**Rating:** 5
**Confidence:** 3

**Summary:**

This paper evaluates encoder-only transformers on a synthetic task, where data is sampled from a complete binary tree-structured generative model of fixed depth $\ell$. There is a knob $k$ that determines at which layer that subtrees are forced to be conditionally independent. This generative process is equivalent to a PCFG, although it is framed mostly in terms of factor graphs and belief propagation. The authors find that transformers can predict the root node type given the leaves with high accuracy. They find that they can do MLM prediction with high accuracy, and that using MLM as a pretraining step improves sample efficiency for root prediction. They analyze attention patterns and see that they attend in a hierarchical fashion as expected.

**Strengths:**

Some of the experimental results are quite interesting. For example, Fig 4 shows the expected attention patterns, and Sec 3.4 is interesting in that it shows that MLM pretraining improves sample efficiency.

**Weaknesses:**

Although the paper includes some interesting visualizations, I am not quite convinced that the contributions of this paper are particularly novel or rise to the level of a full ICLR paper. At times I also found the paper difficult to read, and its claims unclear.

1. In terms of novelty, there seems to be significant overlap with Allen-Zhu & Li (2023), and the experiments seem to be a simpler case of that paper. Like this paper, Allen-Zhu & Li (2023) trained transformers on data generated from CFGs of fixed depth and showed that the the transformer layers learned to attend to constituents as expected.
1. In terms of the significance of the contribution, independently of the question of whether it is novel, this is a very simple synthetic task that seems a bit contrived so that transformers are successful on it (an issue that is also in Allen-Zhu & Li (2023)), and it is not clear that we learn very much about transformers from these experiments. The strings in the training data are all of the same length, and the depth of the underlying parse tree is also fixed and does not exceed the number of transformer layers. It is not surprising that a transformer encoder with the same number of layers as the underlying parse tree can learn to mimic the structure of the underlying complete binary tree. It would be more interesting to test the transformer on a CFG with parse trees of varying depths. Do we see similar behavior, and does the fact that the number of layers is finite matter then?
1. In terms of clarity, it is not clear at the beginning of the paper what its primary goal is. Is the paper primarily about proposing a new data model, and if so, what is the purpose and significance of $k$? Are you primarily interested in analyzing the transformer architecture, and if so, how will the analysis on this synthetic task help us understand the behavior of transformers on real tasks such as natural language?
1. One of the main claims of the paper is that the transformer learns to implement a belief propagation algorithm, but I don't see significant evidence that shows that it is learning to implement BP vs. another algorithm, e.g., some version of the inside algorithm. I don't think the analysis of accuracy on OOD examples and attention patterns rules this case out.
1. Major figures supporting the paper's claims are only in the appendix (Fig 7, App C.3 and C.4).

**Questions:**

1. Intuitively, what "knob" does the filtering parameter $k$ represent? Is it the case that lower $k$ result in more long-range correlations in the data?
1. 035: Another relevant paper: https://arxiv.org/abs/2305.02386
1. Is there a particular reason why you chose to frame the paper mostly in terms of factor graphs and belief propagation, rather than CFGs and standard parsing algorithms (e.g., the inside algorithm)? Is there an advantage to presenting it this way? Is there an advantage in time complexity vs. using a CFG parsing algorithm?
1. 119: What would the equivalent PCFG be, incorporating the depth constraint and $k$?
1. 133: What is $\mathcal{O}_a$? What is $q$? This part is very unclear to me.
1. 144: It's not clear to me what this means. Can you express this in equations?
1. Can the root always be uniquely determined by the input symbols? According to my understanding, the underlying CFG can be ambiguous. How is it possible to get 100% accuracy?
1. How does the BP algorithm described in the main text relate to the experiments? In what way is it used? I don't think this is stated explicitly.
1. Fig 3: Why do you report validation accuracy but not test accuracy? Why does the accuracy go up and then down? Did you not use the best checkpoint when evaluating on OOD data?
1. Why use accuracy instead of perplexity for MLM? Since the CFG can be ambiguous, there isn't only one correct answer, right?

---

> ### Author Response · Authors · 2024-11-19
> **Author response (part 1)**
>
> We thank the reviewer for carefully reading our work and providing valuable feedback, which we will use to improve our presentation. We would first like to address the overall weaknesses identified by the reviewer:
>
> **(W1) Novelty relative to Allen-Zhu & Li (2023):** We understand the reviewer’s concern on this point, especially given the thorough experiments performed in this work on probabilistic CFGs. However, we believe that the objectives of our works are markedly different, albeit complementary. In a nutshell, their work demonstrates that GPTs, which are large and pretrained transformer-based models, have the ability to efficiently parse probabilistic context-free grammars. They notably show this by performing ‘invasive’ probing experiments on these large models, for instance predicting some ancestors in the (variable topology) trees characterizing the samples. However, they do not address (i) what learning dynamics the transformers may follow to achieve this–do they progressively discover the existence of ancestry through training? do they start by combining adjacent symbols through the attention and progressively include the relation between increasingly distant tokens in the sequences?–; (ii) how the trained transformers process the information within the architecture, and the possible relationship between the underlying parsing tree and the neural network; (iii) if the exact inference algorithm, here the inside-outside algorithm for probabilistic CFGs is even implementable in the neural network architecture they are studying, or if the transformers must necessarily rely on some approximation in their implementation due to some architecture related infeasibility. While Allen-Zhu & Li (2023) undoubtedly provide evidence that large transformer-based networks may implement the optimal algorithm for CFGs, we thus believe that they do not tackle the problem from a mechanistic interpretability standpoint. On the other hand, we propose a simplified setting that allows us to specifically tackle points (i)-(iii) stated above, and to gain valuable insights into the inner workings of transformers and how they learn from structured data. Finally, note that the proposed filtering procedure is only possible if tree topology (and the sequence length) is fixed, which is the case for our simplified setting, but not for general CFGs (see also Q3-Q4). Generalizing this tool, which was essential in understanding the sequential discovery of the hierarchical correlations, to the general parsing case is an interesting yet challenging future research direction.
>
> **(W2) Simplicity of the task:** Somewhat related to the point above, we believe that this perceived weakness of our work strongly depends on the question one sets out to answer. We agree with the referee that it is indeed not a surprise in itself that, given enough samples, transformers manage to fit the training data distribution–after all, we know that actual language is perfectly mastered by LLMs. However, our work sets out to answer the question of how transformers are able to assimilate complex probability distributions, both in the sense of the training dynamics as well as the ‘spatial’ organization of trained transformers and how they perform computations throughout the architecture. In this context, we believe that our setting provides an intermediate complexity between probabilistic CFGs (see the point above related to Allen-Zhu & Li (2023)), and the typical state of the art mechanistic interpretability settings that often rely on simple mathematical tasks such as modular addition (Zhong et al., 2024) or else histogram counting (Behrens et al., 2024). As also mentioned in the answer to the previous point, the fixed depth allows us to introduce the filtering procedure, which is a key interpretative element in our analysis.
>
> **(W3) Clarification of the goals of the paper:** We thank the reviewer for motivating us to restructure our abstract and introduction, which did not carry the goals and messages of our paper properly. In the revised version, we aim to recenter the introduction towards the title of the paper and our overarching goal of understanding how structure may be digested in transformer architectures and notably in the attention mechanism. The filtered data model is therefore a tool that allows us to achieve this goal in the context of well-controlled inference problems. We believe that applications of our insights towards practical problems, in NLP or other, are tangible. We added a discussion on the fact that the structure of data and wide-spanning correlations being progressively incorporated by the transformer during training is a feature that has been found in widely different contexts, and that may also have valuable implications for the practical design of training protocols.

---

> ### Author Response · Authors · 2024-11-19
> **Author response (part 2)**
>
> **(W4) Evidence of BP vs other algorithms:** We appreciate this concern of the reviewer, and admit that further evidence, which we now provide in the updated version, allows us to more convincingly support our claim. Indeed, we have added in the revised version of our manuscript a more precise metric in the form of the Kullback-Leibler divergence between the BP marginals and the softmax (instead of argmax used for prediction) of the neural network outputs. We find that this quantity decreases following a staircase along the factorization levels during training, and eventually reaches small values in fully trained networks, showing that the ‘full’ (per-sample) output of the networks and BP match, and not just their accuracies. Nonetheless, we would like to stress several points that were perhaps insufficiently clear in the manuscript. In our data model, Belief Propagation is not simply an effective algorithm, it is the information-theoretically optimal oracle for any symbol inference task, whether it be the root or the leaves. As a result, matching its performance both in-sample (tested against the model it was trained on) and out-of-sample (tested against larger filtering levels) is a strong clue that the transformer implementation could be a close approximation of the exact algorithm. Moreover, it may be shown that the inside-outside algorithm is in fact identical to BP (Sato, 2007), when the topology of the parsing tree is given. Generic probabilistic CFGs may indeed be seen as tree-based graphical models with an additional probability distribution over the realization (i.e., topology) of the parsing tree. Finally, we would like to point out that the alignment between the output of the transformer and the optimal inference algorithm constitutes only part of the presented evidence. It has to be understood in conjunction with the ‘spatial’ organization of the attention maps, the interpretation of which is verified by our probing experiment at higher ancestry levels; this organization is compatible with the computational graph of BP. We feel that we have not managed to relate these points together convincingly in the current version of the manuscript, and we will address this issue by adding a comprehensive summary of our results, to paint a more complete picture of our analysis.
>
> **(W5) Major figures in the appendix:** Unfortunately, we cannot easily comply with this request by presenting all our evidence in the main text, given the stringent page requirement of ICLR. On the other hand, we believe that the results presented in Fig. 7 and the content of Appendices C.3 and C.4 provide secondary evidence in support of our claims. Fig. 7, for instance, shows that also transformers trained on filtered data models can reach optimal inference performance, which in our opinion is mostly a sanity check (optimal inference is already reached, as shown for the full data model, in Fig. 1). In a similar vein, Appendices C.3 and C.4 are consistent with our findings but do not lead to any independent conclusions.

---

> ### Author Response · Authors · 2024-11-19
> **Author response (part 3)**
>
> To address the referee’s questions:
>
> **(Q1) What "knob" does the filtering parameter k represent?** In simple words, k>0 creates a shortcut in the hierarchy of the generative tree (see Fig. 1). When we filter out the upper levels, the root directly generates 2^(k) ancestors, independently one from the other. This means that strong correlations, induced by the remaining (l-k) unfiltered layers, survive only within blocks of size 2^(l-k). Therefore, (as suggested by the reviewer) the lower the parameter k, the stronger the longer-range correlations in the data.
>
> **(Q2) Other relevant paper Khalighinejad et al. (2023).** We were not aware of this work, indeed related to transformers on probabilistic CFGs. We thank the referee for bringing it to our attention, the reference was added in our introduction. Similarly to Allen-Zhu & Li (2023) and Zhao et al. (2023), the paper studies the ability of transformers to reconstruct parsing trees, but does not address (i) the learning dynamics, (ii) the implementation of the algorithm within the architecture and (iii) the existence of an exact transformer based implementation of the classic algorithm. This additional reference reinforces our interest in interpreting algorithm discovery within transformer architectures.
>
> **(Q3) Graphical models & BP vs CFG parsing algorithms.** This is an interesting point, which we can answer along two main motivations. The first is simplicity and tunability. As mentioned above, we set out to go beyond the existing works on CFGs on the question of mechanistic interpretation and to understand more precisely how an inference algorithm is learned and implemented in transformers. This required a simplification of the data model, and notably relying on a fixed topology, allowing us to introduce the filtering parameter–i.e. to tune the range of correlations in sequences. In this setting, the standard CFG terminology would be misleading, as our data model lacks the central feature of probabilistic CFGs: variable sequence length and the related existence of terminal and non-terminal symbols. The other motivation is that our setting is slightly more model agnostic. Indeed, while not tuned to describe natural language, probabilistic graphical models can describe many other objects, from protein sequences to random graphs. In this respect, our work could be used to make contact with markedly different problems where transformers could also be effective, for instance community detection in stochastic block models (Mossel et al., 2014) or planted graph coloring problems on graphs (Krzakala & Zdeborova, 2009), for which Belief Propagation is still an effective algorithm.
>
> **(Q4) What would the equivalent PCFG be, incorporating the depth constraint and k?** As already mentioned in W4 and Q3, it is unclear that a standard PCFG could be formulated to be equivalent to our model, since it would require one to remove the intrinsic ambiguity related to the topology of the parsing tree. Only in a fixed topology case (implying a fixed sequence length), the hierarchical tree can be cut at a specific level with the filtering procedure. In fact, in a general PCFG, the terminal symbols could be found at any level of the parsing tree, and their correlations to the other leaves is to be determined at inference time. In our model, the correlation between the elements of the sequence (i.e. the terminal symbols), is fully specified by the positions in the sequence. Therefore, generalizing the hierarchical filtering procedure to PCFGs is non-trivial. Moreover, we are not considering a separate dictionary for the non-terminal symbols and for the root symbol. While this choice may make little sense from the perspective of linguistics, it allows us to define a non-trivial root classification task, which would not be possible in a normal PCFG.

---

> ### Author Response · Authors · 2024-11-19
> **Author response (part 4)**
>
> **(Q5) What is $O_a$? What is $q$?** We start by acknowledging that our presentation was not effective, we will rewrite this paragraph for clarity. To address the question: $q$ is the size of the dictionary of symbols that can be taken by any element of the sequence, and by any node on the ancestry tree starting from the root (see Q4). $O_a$ represented the set of possible children pairs generated by a parent symbol $a$ (i.e. the allowed production rules). What we mean by $O_a \cap O_{a’} = \emptyset$ is that the production rules we consider are completely distinct for each parent symbol, making our unfiltered model non-ambiguous (at odds with more general PCFGs). Given a pair of children, there can only be one possible parent in the layer above. $|\cup_a O_a |=q^2$ on the other hand means that any possible children pair can be produced, and the rules are equal partitioned among the $q$ possible parent values. To summarize, our model is described by a transition tensor that has $q\times q \times q$ entries. It is then organized as q ‘slices’ populated by $q$ entries (the rest of the entries being zero), which are non-overlapping for each slice.
>
> **(Q6) l.144: It's not clear to me what this means. Can you express this in equations?** This is related to the point above. What we mean is that in the unfiltered cases, transitions in the tree are of the form $M(a \to bc)$ where a is the parent symbol and b and c are the children, as shown by black factors in Fig.1a. By strongly correlated we mean that in each transition b and c are not drawn independently but as a pair. We hope this clarifies our statement.
>
> **(Q7) Can the root always be uniquely determined by the input symbols?** Yes, this is the case with our choice of non-ambiguous production rules in the unfiltered (full hierarchical) model (see Q5). Given a pair of children symbols, one can exactly infer their parent. Combining this for all pairs in the sequence and going up the tree, the root can be determined with certainty. This difference with PCFGs is central, and justifies our framing in terms of a probability distribution described by a factor graph rather than in terms of CFGs, see Q3.
>
> **(Q8) How does the BP algorithm described in the main text relate to the experiments?** Given any filtered/unfiltered hierarchical model, one can derive the associated BP algorithm (using explicit knowledge of the production rules and the corresponding transition rates), which represents the exact oracle for any inference task defined in the context of the graphical model. As such, it provides an information-theoretic bound for the accuracy of the reconstruction of hidden symbols (e.g, the root, or the masked tokens). This is what is shown by the black dashed line in Figs. 1b), 1c), 1d), Fig. 3 and Fig. 6; and by all dashed lines in Fig. 7. It also provides a comparison point in out-of-sample tests, i.e. used on data generated from different levels of filtering compared to the training data, which is shown by the colored lines on Figs. 1b), 1c), Fig. 3 and Fig. 6. As stated in the point above, the accuracy is trivially equal to 1 for the root classification given an entire sequence for fully hierarchical data.

---

> ### Author Response · Authors · 2024-11-19
> **Author response (part 5)**
>
> **(Q9) Why do you report validation accuracy but not test accuracy?** We apologize for the abuse of terminology on our part. We do not have distinct test and validation sets in our experiments. The models are trained on a training set for a fixed amount of epochs, and we simply measure the test accuracy for the final configuration of model weights (not the best). We corrected this mistake by replacing validation accuracy with test accuracy throughout the paper. **Why does the accuracy go up and then down?** This behavior is very important for the interpretation of the learning dynamics. We will clarify this point in the revised version, given that the current presentation was ineffective. In a nutshell, the accuracy goes up and down on test sets generated from different (higher) filtering data than the training set, and therefore OOD from the perspective of the training model. The explanation comes from the fact that the transformer discovers the existence of higher hierarchical correlation levels (i.e., longer-range correlations) sequentially, during training. It starts by imputing a simplistic explanation for the data (e.g., a correlation length of 2), thus initially increasing its accuracy on the corresponding OOD data. But then an additional correlation level (e.g., correlation length of 4) is imputed, and the transformer stirs its predictions accordingly. At this point, the accuracy on the simpler, more factorized data decreases, since the model is learning to assume a richer correlation structure between the symbols, which is not present in the simpler OOD data. This staircase behavior is novel in our work relative to all the PCGF publications mentioned above. We believe that it may be a more general feature of deep network learning dynamics (Refinetti et al., 2023; Bardone & Goldt, 2024; Rende et al., 2024). **Did you not use the best checkpoint when evaluating on OOD data?** As stated above, we used the model weights obtained at the end of training. Our objective is to understand the inner workings of the learning, not to achieve the best possible performance on the OOD data.
>
> **(Q10) Why use accuracy instead of perplexity for MLM? Since the CFG can be ambiguous, there isn't only one correct answer, right?** This is a fair remark, which was also indirectly raised by reviewer c8Kw. We have taken it into account in our revised version, including new experiments that we believe strengthen our evidence. We measured the Kullback-Leibler divergence between the BP marginals and the softmax (instead of argmax for a prediction) of the transformer output. This is not exactly the perplexity, however, we believe that it is more relevant for our point of understanding how the networks perform inference while taking into account the necessarily ambiguous nature of the prediction. We find that the KL divergence decreases in training following the same staircase scenario on the filtered models as described in Q9, and show that the full probabilistic prediction matches the exact one given by BP, and not only the accuracy.
>
> **References:**
>
> Allen-Zhu, Z., & Li, Y. (2023). Physics of language models: Part 1, context-free grammar. arXiv preprint arXiv:2305.13673.
>
> Bardone, L., & Goldt, S. (2024). Sliding down the stairs: how correlated latent variables accelerate learning with neural networks. arXiv preprint arXiv:2404.08602.
>
> Behrens, F., Biggio, L., & Zdeborová, L. (2024). Understanding counting in small transformers: The interplay between attention and feed-forward layers. In ICML 2024 Workshop on Mechanistic Interpretability.
>
> Khalighinejad, G., Liu, O., & Wiseman, S. (2023). Approximating CKY with Transformers. arXiv preprint arXiv:2305.02386.
>
> Krzakala, F., & Zdeborová, L. (2009). Hiding quiet solutions in random constraint satisfaction problems. Physical review letters, 102(23), 238701.
>
> Mossel, E., Neeman, J., & Sly, A. (2014). Belief propagation, robust reconstruction and optimal recovery of block models. In Conference on Learning Theory (pp. 356-370). PMLR.
>
> Refinetti, M., Ingrosso, A., & Goldt, S. (2023). Neural networks trained with SGD learn distributions of increasing complexity. In International Conference on Machine Learning (pp. 28843-28863). PMLR.
>
> Rende, R., Gerace, F., Laio, A., & Goldt, S. (2024). A distributional simplicity bias in the learning dynamics of transformers. arXiv preprint arXiv:2410.19637.
>
> Sato, T. (2007). Inside-Outside Probability Computation for Belief Propagation. In IJCAI (pp. 2605-2610).
>
> Zhao, H., Panigrahi, A., Ge, R., & Arora, S. (2023). Do transformers parse while predicting the masked word?. arXiv preprint arXiv:2303.08117.
>
> Zhong, Z., Liu, Z., Tegmark, M., & Andreas, J. (2024). The clock and the pizza: Two stories in mechanistic explanation of neural networks. Advances in Neural Information Processing Systems, 36.

---

> > ### Comment · Reviewer_VwvV · 2024-11-25
> >
> > Thank you for your responses. I appreciate that you have taken the time to clarify the goals of paper and revise the draft.
> >
> > So, as I understand it, the thesis of the paper now is that transformers learn to model hierarchies in bottom-up order throughout training, and the data model was designed in service of testing that hypothesis. Is that a fair characterization? Are there other contributions that should be highlighted?
> >
> > If I may push back on your point about expressing the data model as a PCFG -- I think it would be extremely helpful to many readers (including me) to give an equivalent PCFG. I'm not asking how you would translate an arbitrary PCFG to your data model, just how you would express your data model as a PCFG for a given $k$. If it isn't possible, then the remarks about equivalence to PCFGs should be removed.
> >
> > Quick question: What is the time complexity of the BP algorithm?

---

> ### Author Response · Authors · 2024-11-25
> **Re:**
>
> We thank the referee for taking the time to go through our response and engage in this dialogue.
>
> **On our contributions**
>
> Our original goal, which we hope is more clearly stated in the revised version, is to understand the learning process and the computation of a transformer trained on our data in a mechanistic way, in the light of the exact, known inference algorithm. To achieve that, on top of the point highlighted by the reviewer,
> * We show that the transformer reaches calibration, i.e. approximates the exact output of the inference oracle, even on OOD data.
> * We analyze how the exact computation is embedded in the transformer weights (learning “in space”).
> * We provide a novel feasible implementation of the exact algorithm within the same architecture and show that probing experiments on the trained transformer qualitatively align with some of its properties.
>
> **Equivalent PCFG formulation**
>
> A probabilistic context-free grammar G can be defined by a quintuple $G=\left(M,T,R,S,P\right)$, where $M$ is the set of non-terminal symbols, $T$ is the set of terminal symbols, $R$ is the set of production rules, $S$ is the start symbol, $P$ is the set of probabilities on production rules.
>
> In our model, we consider the special case where terminals and non-terminals coincide $M=T\overset{def}{=}\mathcal{S}$, and there is a set of possible root symbols $\mathcal{R}$. We pick $|\mathcal{R}|=|\mathcal{S}|=q$. The production rules $R$ and their probabilities $P$ are defined as:
> * For $k>0$: $\forall r\in\mathcal{R}$, all the trasitions of the type: $r\rightarrow s_{1}...s_{2^{k}}$ are allowed, with probability
> $P\left(r\rightarrow s_{1}...s_{2^{k}}\right)=\prod_{k}P^{(k)}\left(s_{k}|r\right)$, computed as in Eq. (1) in the revised paper. Moreover, $\forall s\in\mathcal{S}$, we allow $q$ transitions of the type: $s\rightarrow s_{1}s_{2}$ with probability $P\left(s\rightarrow s_{1}s_{2}\right)=M_{ss_{1}s_{2}}$.
> * For $k=0$: the production rules for the root correspond to those among the symbols: $r\rightarrow s_{1}s_{2}$ with probability $P\left(r\rightarrow s_{1}s_{2}\right)=M_{rs_{1}s_{2}}$.
> * For $k=\ell$: we only have root-to-leaves production rules of the type: $r\rightarrow s_{1}...s_{2^{\ell}}$, with probability $P\left(r\rightarrow s_{1}...s_{2^{\ell}}\right)=\prod_{k}P^{(k)}\left(s_{k}|r\right)$.
>
> Moreover note that, since we impose a non-ambiguity constraint over the transitions, we have that $\forall$ children pair $s_1,s_2$, $M_{p(s_1,s_2),s_1,s_2}\neq 0$ for a single parent symbol $p(s_1,s_2)$, i.e. any pair of children symbols can come from only one parent symbol.
>
> Finally, we consider a fixed parsing tree: a full tree with $2^\ell$ leaves.
>
> **Question on BP**
>
> To answer the referee’s question, the time complexity of the BP algorithm on a tree is linear in the sequence length $n=2^\ell$, or exponential in the tree depth, $\mathcal{O}(n)$. Indeed, there is a large computational discount from knowing the topology of the parsing tree, compared to the inside-outside algorithm (which instead scales as $\mathcal{O}(n^3)$).

---

> > ### Comment · Reviewer_VwvV · 2024-11-25
> >
> > Thanks for the clarifications! A lot of my concerns about clarity and the motivations of the paper have been answered, so I'm willing to raise my score from a 3 to a 5. The difference in contribution between Allen-Zhu & Li (2023) and this paper has also been made clearer, although I would push back a bit on claim (ii) -- they argue at some length that the transformer implements a dynamic programming algorithm, perhaps akin to CKY or the inside algorithm, albeit their argumentation on this point isn't very clear. In my opinion, the scope of the contributions of the paper are below the acceptance threshold for ICLR, which is why I have not raised it to a 6.
> >
> > I do have some advice for strengthening the contributions of the paper.
> >
> > > We provide a novel feasible implementation of the exact algorithm within the same architecture and show that probing experiments on the trained transformer qualitatively align with some of its properties.
> >
> > > Indeed, there is a large computational discount from knowing the topology of the parsing tree, compared to the inside-outside algorithm
> >
> > These observations point to some interesting questions. Allen-Zhu & Li (2023) argued that the transformer learns to implement a DP algorithm related to CKY or inside-outside. This claim is basically wrong, because those algorithms require $O(n^3)$ time, but the transformer runs in only $O(n^2)$ time. On the other hand, your paper gives an instance of a hierarchical pattern that can be processed in $O(n)$ time. We also know from Khalighinejad et al. (2023) that transformers can approximate CKY to some extent in $O(n^2)$ time. Your BP interpretation could help to explain this gap. What is the class of PCFG that can be learned? You mentioned that knowing the topology of the tree is helpful, but the structural information is not explicitly given to the transformer -- it still needs to learn it. I think what your work shows is that once the transformer learns the topology, then it can compute marginals efficiently. Explaining the findings of Allen-Zhu & Li (2023) and Khalighinejad et al. (2023) in terms of the learnability of tree topologies and BP would be an interesting line of inquiry.

---

> > > ### Author Response · Authors · 2024-11-26
> > > **Re:**
> > >
> > > We thank the reviewer for acknowledging the role our work could play in understanding which parsing-related inference algorithms can be feasibly approximated with a transformer, and for suggesting a direction they believe could be worth exploring. At the moment, as mentioned in our conclusions, we are indeed continuing our research in the direction of reintroducing the variability of the parsing tree topology, and hope to make progress in understanding in a detailed fashion when and how the transformer can still approximate the exact output. We agree with the reviewer that our work provides a strong baseline for these further explorations.
> > >
> > > On the other hand, we invite the reviewer to consider also the other contributions we are providing, although they might not align with the research questions they find most interesting. We are contributing to different lines of research:
> > > * Finding exact embeddings of inference algorithms in neural network computation: e.g., [1] does so without any experimental evidence, [2] proposes an unfeasible scaling for the network parameters.
> > > * Understanding which inference problems related to parsing can be solved approximately optimally: e.g., the reviewer argues that [3] claims cannot be correct, and [4] shows that the approximation degrades with high ambiguity.
> > > * Mechanistic interpretation of the transformer computation: e.g. [5, 6, 7] attempt to interpret the computation in tasks where the algorithms for obtaining the correct answer are available, through attention map analyses and probing experiments.
> > > * Understanding the role of the learning dynamics, and how different components of the data correlation structure are discovered in time: e.g., [8, 9] show similar stair-case phenomenologies, but with different data models.
> > > * Understanding the role of hierarchical correlations: e.g. [10, 11] show how they are absorbed and how they shape the learning of deep networks.
> > >
> > > Many of these works have been accepted at major conferences. In our work, we strive to propose new evidence and make progress in each of these research directions. However, the reviewer believes the content presented in our work is **not** even **weakly acceptable** in ICLR. Given that the content and the claims of the paper have been accepted by the reviewer, we can only hope they can reconsider their judgment.
> > >
> > > **Bibliography**
> > >
> > > [1] Song Mei, “U-Nets as Belief Propagation: Efficient Classification, Denoising, and Diffusion in Generative Hierarchical Models
> > >
> > > [2] Zhou et al. “Do Transformers Parse while Predicting the Masked Word?”
> > >
> > > [3] Allen-Zhu et al. "Physics of language models: Part 1, context-free grammar."
> > >
> > > [4] Khalighinejad et al. "Approximating CKY with Transformers."
> > >
> > > [5] Rende, et al. "Mapping of attention mechanisms to a generalized Potts model."
> > >
> > > [6] Behrens et al. "Understanding counting in small transformers: The interplay between attention and feed-forward layers."
> > >
> > > [7] Zhong et al. "The clock and the pizza: Two stories in mechanistic explanation of neural networks."
> > >
> > > [8] Bardone, et al. "Sliding down the stairs: how correlated latent variables accelerate learning with neural networks."
> > >
> > > [9] Székely et al. "Learning from higher-order statistics, efficiently: hypothesis tests, random features, and neural networks."
> > >
> > > [10] Cagnetta et al. "How deep neural networks learn compositional data: The random hierarchy model."
> > >
> > > [11] Cagnetta et al. "Towards a theory of how the structure of language is acquired by deep neural networks."

---

### Official Review · Reviewer_c8Kw · 2024-11-03

**Soundness:** 2
**Presentation:** 3
**Contribution:** 2
**Rating:** 5
**Confidence:** 3

**Summary:**

The paper investigates how transformer models make predictions on samples coming from a structured data distribution, focusing on the hypothesis that transformers implement belief propagation to make predictions.

Contributions:
1. The authors propose a novel family of synthetic data distributions based on PCFGs to test the hypothesis empirically.
2. The authors present experimental results on two tasks, root prediction and masked language modeling, which the authors claim to support their hypothesis.
3. The authors present a construction for how to implement belief propagation for a tree-structured factor graph of depth $l$, using only $l$ layers.

**Strengths:**

1. The work is well-placed in the context of other mechanistic interpretability work for transformers, as well as other work looking into the effect of structured data on machine learning models.
2. Understanding what strategy is learned by transformer models trained on structured distributions is an important problem.
3. The family of synthetic distributions is interesting and has a hyperparameter that allows one to control the locality of correlations between tokens in the sequence.
4. The authors present a novel construction for how to implement BP on a depth $l$ tree using only $l$ layers of a transformer, whereas previous constructions required $2l$ layers.

**Weaknesses:**

1. It’s not clear how much the observations made in this work generalizes to larger models, and more complicated data distributions.
2. The empirical evidence presented has alternative interpretations that haven't been ruled out. (Please see Questions.)
3. The construction for implementing BP on depth-$l$ tree using a transformer of only $l$-layers could benefit from a bit more details, especially on the root-to-leaf message passing part. (Please see Questions.)

**Questions:**

W2.1: In both the supervised root prediction task and the MLM task, the authors argue that the transformer performs similar in accuracy to BP is evidence that the transformer is implementing an approximation to BP. While it is very intriguing that the accuracies are so systematically similar, there are plausible alternative explanations (that additional experiments or analysis could rule out):

W2.1.1: Similar accuracy doesn’t imply similar behavior on individual inputs. Have authors considered measuring the match between BP and transformer predictions on individual inputs? If the match is high, this would strengthen the authors’ claim that the model actually behaves like BP.

W2.1.2: Similar behavior on individual inputs doesn’t imply similar implementation. For example, in figure 1(c), the authors show accuracy of a model trained on k=0 (and for root classification) data and evaluated on k>=0 data, whose accuracy is similar to running BP on the k=0 graph. The authors claim this is evidence in support of model having learned to implement BP. Why couldn’t the model have model the k=0 data well without implementing BP? BP on k=0 graph is optimal for k=0 data, so the transformer that was trained on lots of k=0 data would necessarily behave like BP on individual inputs, which (without necessarily implementing BP) would make similar predictions to BP on out-of-sample data as well.

W2.2 In the supervised root prediction task, the authors write that (line 314-316) “We interpret this as a consequence of the weaker correlations between distant tokens—and therefore the lower signal-to-noise ratio during learning—that must be resolved to match the BP prediction.” Since the task is to predict the root, could the authors elaborate more on why they believe the weaker correlations among **tokens within a sequence** is causing difficulty for learning? An alternative interpretation is that this is due to the weaker correlation between the **root** and the **entire sequence** of tokens.

W3. Following up on the BP construction, could the authors elaborate more on the leaf-to-root passing? In particular, what do the $r$’s represent? It looks like all $r$’s are initialized to uniform distribution, and they each get updated with the same formula (eqn 22), so would $r^{(a,m)}_i$ ever be different from $r^{(a’,m)}_i$ for $a \neq a’$? A walkthrough of the formulas on a minimal example with small $l$, and $q$ may be helpful here.

Other: Did the authors mean to refer to figure 3 instead of 1c on lines 337? In the caption of figure 1c it says it’s for MLM instead of root classification, and the scale of the x-axis suggests it’s MLM too.

---

> ### Author Response · Authors · 2024-11-19
> **Author response (part 1)**
>
> We thank the reviewer for carefully reading our work and providing valuable feedback, which we will use to improve our presentation. We would first like to address the overall weaknesses identified by the reviewer:
>
> **(W1) It’s not clear how much the observations made in this work generalizes to larger models, and more complicated data distribution:** Considering somewhat prototypical tasks is customary and often necessary when the goal is to achieve a mechanistic interpretation of the transformer’s computation (Zhong et al., 2024; Behrens et al., 2024), and we believe it can still provide intuitive explanations that generalize to more complicated cases. For example, in our work, we can argue that the sequential discovery of higher levels of hierarchical correlations, involving longer-range token interactions, is likely to shape learning processes also in real-world NLP tasks. We clarified this point in the revised version.
>
> **(W2) The empirical evidence presented has alternative interpretations that haven't been ruled out:** We thank the reviewer for motivating us to follow up on their question with some new experiments, which will be included in the revised version. In particular, we compare the full output probability distributions of the transformer and BP (via a KL divergence in the main text, and the Spearman correlation coefficient and scatter plots in the appendix), as a function of the training epochs. Not only do we find that the model learns to approximate the exact BP marginals, but we also show that, at intermediate stages of learning, the transformer sequentially aligns its predictions with the filtered-BP marginals, recovering the same stair-case behavior observed in Fig. 1(c). This evidence reinforces our picture of the consecutive discovery of hierarchical correlation levels in transformers. We address the reviewer’s more specific questions on this topic below.
>
> **(W3) The construction for implementing BP on depth-l tree using a transformer of only l-layers could benefit from a bit more details, especially on the root-to-leaf message passing part:** We thank the reviewer for reading carefully our appendix and taking interest in this more challenging part of our work. We agree that the current explanation is insufficient to fully understand the role of “r”. We added a new paragraph, showing how the “r” recursion is obtained from the standard BP recursion, by conveniently playing with the traced indices. We address the well-spotted missing part of our recursion on “r” in the answer to the questions Q3 below.

---

> ### Author Response · Authors · 2024-11-19
> **Author response (part 2)**
>
> To address the referee’s questions:
>
> **(Q1) Have authors considered measuring the match between BP and transformer predictions on individual inputs?** Again, we thank the reviewer for encouraging us to push the comparison further. Please see answer to W2.
>
> **(Q2) Similar behavior […] doesn’t imply similar implementation. Why couldn’t [...] transformers […] behave like BP on individual inputs […] on out-of-sample data as well […] without necessarily implementing BP?** We thank the reviewer for this key criticism, which was similarly raised by reviewer VwvV. Given the relevance of this point in the present work, we decided to substantially rewrite the related paragraphs, clarifying what we meant by “learning an implementation of BP” (now rephrased as “learning to approximate the exact inference computation”) and how we reached this conclusion. Our claims are based on the following evidence:
> - We obtain approximately identical outputs (in root prediction and MLM) from equal inputs, both in- and out-of-sample. Note that the model is “trained to match BP” only indirectly (we train on the correct values of the masked symbols, not on reproducing the BP marginals, which could differ) and on the training distribution. If the match was accidental, and purely driven by data-fitting, the predictions would differ in the OOD case. For our purposes, obtaining the same output on any sequence implies an equivalence in the computation.
> - We observe the same hierarchical computational structure underlying BP arises in the transformer (see attention maps), with a sequential focus on longer correlation lengths, corresponding to the various levels of the tree. Indeed, the transformer just learns to “model the data well”. But in this case, it does this so well that the correct sequence of operations (underlying the exact inference procedure) is discovered. The computation doesn't need to be done precisely in the same way as in a standard BP implementation (in fact, it is embedded in high-dimensional space, and different combinations of non-linear and linear operations are employed). Still, there is a fundamental equivalence in how single token information is collected and integrated to eventually obtain the exact predictions for the masked tokens.
> - Performing probing experiments on ‘ablated’ transformers, we confirm what was hinted by attention maps, that is that removing the k last attention blocks in the architecture leads a transformer that was trained on the full data model to achieve a similar performance at predicting the l-kth ancestor as one that was trained on a data model with a level of filtration k. This strongly reinforces our claim that the transformer goes up the generative tree as the tokens are passed through its architecture, as it would do if following our tentative transformer-based implementation of BP in l layers.

---

> ### Author Response · Authors · 2024-11-19
> **Author response (part 3)**
>
> **(Q3) Since the task is to predict the root, could the authors elaborate more on why they believe the weaker correlations among tokens within a sequence is causing difficulty for learning?** We thank the reviewer for raising this question, as it pushed us to look a bit more attentively at this issue and to provide a new (simpler) interpretation that will hopefully convince them as well. To answer their question on why we thought this was related to token-token correlations rather than sequence-root correlations, this is because the k=l case is actually the one which has the smallest mutual information between the sequence and the root, while it is the easiest to learn from Fig.7 in the appendix. Moreover, this mutual information should be strictly increasing as k decreases, while the sample complexity as a function of k appears to present a non-monotonic behavior, being the smallest for k=l and the second smallest for k=0, therefore we ruled out the correlations between the root and the entire sequence as the cause of this phenomenon. However we missed an important aspect of the training, which is singular to the k=0 case. The deterministic nature of the root inference problem for k=0 makes it unique for two reasons: the first is that the logits outputted from the network need not be calibrated, so the accuracy can reach the optimum without the transformer having fully implemented an algorithm equivalent to BP, whereas the relative weights of prediction must be well understood to match the optimal inference in the ambiguous k>0 cases, this is visible on Fig.7; the other is that this being said, matching the BP is also easier in the k=0 case because the training  cross-entropy loss corresponds exactly to that computed with the true BP marginals (that are also delta distributed due to the determinism again) whereas in the k>0 cases the training loss does not guide explicitly to the BP marginals, this is visible by re-plotting Fig.7 with the Kullback-Leibler divergence between the network logits and the BP marginals instead of the accuracy. Bringing these two points together, we clearly understand why the k=0 case requires less samples than intermediate k. At the other end of the spectrum, it is understandable that the k=l case is the easiest, as it is implementable in a single feedforward layer, as it requires only a Naive Bayes classifier and not an implementation equivalent to full BP. The intermediate cases then appear more or less equivalent, as they require an implementation of BP while not being guided towards the correct marginals during training and not benefitting from the argmax which washes away approximations of the correct marginals in the accuracy.
>
> **(Q4) [...] what do the r’s represent? It looks like all r’s are initialized to uniform distribution, and they each get updated with the same formula (eqn 22), so would ri(a,m) ever be different from ri(a′,m) for a≠a’?** We thank the reviewer for reading carefully our appendix, and spotting the absence of the base case of the recursion for the “r” messages (which we are now reporting). Moreover, we agree that the current explanation is insufficient to fully understand the role of “r”. We added a new paragraph, showing how the “r” recursion is obtained from the standard BP recursion, by conveniently playing with the traced indices.
>
> **(Q5) Refer to figure 3 instead of 1c on lines 337**: We thank the reviewer for spotting this mistake. However, we meant to refer to figure 1b. We fixed this typo in the revised version.
>
> **References:**
>
> Behrens, F., Biggio, L., & Zdeborová, L. (2024). Understanding counting in small transformers: The interplay between attention and feed-forward layers. In ICML 2024 Workshop on Mechanistic Interpretability.
>
> Zhong, Z., Liu, Z., Tegmark, M., & Andreas, J. (2024). The clock and the pizza: Two stories in mechanistic explanation of neural networks. Advances in Neural Information Processing Systems, 36.

---

> ### Comment · Area_Chair_Xhyf · 2024-11-25
> **Please discuss further**
>
> Have the authors adequately addressed your concerns about generalizability and rigor? Are the presentation changes enough to adjust your score? Rebuttal period is ending soon.

---

> > ### Comment · Reviewer_c8Kw · 2024-11-25
> > **Thank you for the updated draft and your response. Some additional clarifications and concerns.**
> >
> > I thank the authors for their responses, the updated draft, and the additional results. I very much appreciate the additional clarifications in the appendix on the construction of BP downward pass.
> >
> > If I understood correctly, the claims made by this updated version is as follows:
> >
> > On learning process:
> > 1. Transformers learn local dependencies before learning longer range ones.
> >
> > On embedded computation:
> >
> > 2. It approximates BP at the input-output level (i.e. matches the distribution, end-to-end).
> > 3. Furthermore, it matches BP at the input-output level because it implements something like BP internally.
> >
> > I appreciate the authors for strengthening their analysis in support of 1  (e.g. Fig 1c and 1d, Fig 4 and 5) and 2 (e.g. Fig 1b). My main concern however (see question W2.1.2 in original review), is still with claim 3, which is central to the contribution of this work.
> >
> > C1. I would first like to clarify with authors whether they are arguing for claim 3 or claim 2 on lines 314-315, 331-332, and 390-392 (my current interpretation of the writing is for claim 3, but I do not believe the evidence is strong enough for 3, since evidence is only at the input-output level). My point is that a model that solves the tasks in section 3 without BP will also yield the observed patterns such as those in figure 1b, 1c, 1d, and 4, 5. I am happy to expand and discuss my reasoning about this further with the authors, but will for now focus the more significant concern I have in C2.
> >
> > C2. If I take the current position that the lines above only support claim 2, then is it fair to say that the main argument for claim 3 relies on results from the probing experiment in section 4 (Fig 7 left)?
> >
> > There’s not enough details in the current paper to form a judgement about the validity of the argument made with Fig 7 left. Here are some important details:
> >
> > How much training data does the probe require to perform like Fig 7 left? And how did you make sure that the probe is not overfitting the training data (i.e. that the marginals about ancestors are available in the representation of the transformer, and easily decoded by the probe, rather than learned by the probe from lots of training data)? For example, in the cited paper Zhao et al. 2023, section 4.3, their probing results involves training a probe on PCFG data and transferring the probe without much loss of accuracy to PTB data.
> >
> > To summarize, my main concerns are as follows:
> >
> > Claim 3 is central to the paper, yet
> > 1. Some arguments seem to be made about Claim 3 using results that I believe is only enough to support claim 2 (C1). (If I misunderstood, and that the authors are only arguing for claim 2 then I don’t have this concern anymore).
> > 2. The most important direct evidence for Claim 3 is section 4 probing, but the details of the probing experiments are not enough to judge whether the probe has overfitted and whether traces of possible BP computation is actually in the transformer activations. (C2)
> >
> > Haoyu Zhao, Abhishek Panigrahi, Rong Ge, and Sanjeev Arora. Do transformers parse while predicting the masked word? arXiv preprint arXiv:2303.08117, 2023.

---

> > > ### Author Response · Authors · 2024-11-27
> > > **Re 2 (1/2):**
> > >
> > > It is clear that there is a misunderstanding about the statements on **algorithmic equivalence**, and that we might have a stronger opinion on the explanation of the computational mechanism implemented by the transformer compared to the reviewer. We acknowledge that we are not proving that BP is exactly implemented by the transformer. However, we think we are not overstating this point in the paper, as we will clarify below. Below, we address the more technical points raised by the reviewer in his latest response.
> > >
> > > **About C1.**
> > >
> > > To better specify our claims, in the **Our contributions** paragraph we claim:
> > >
> > > (reviewer’s **point 2**) *Transformers approach optimal performance … in a calibrated way, by predicting probabilities that approximate those yielded by the BP oracle even on out-of-sample inputs … evidence of equivalence in computation to the exact inference algorithm.*
> > >
> > > (reviewer’s **point 3**) *We find that the attention maps are compatible with a natural
> > > implementation of BP within the architecture, see Fig. 1(e). We verify this **affinity** through
> > > probing experiments, providing strong clues on how transformers learn from our structured data in “space”.*
> > >
> > > The evidence the reviewer points to (the referenced lines) explicitly supports **point 2**, i.e. that the transformer approximates BP at the input-output level (matches the distribution, end-to-end also on random inputs). The **computational equivalence** we claim is at the level of input-output association (please indicate a better phrasing if this sentence appears too strong).
> > >
> > > The coincidence between the mappings provides a motivation for our mechanistic interpretation study, where we try and open up the computation of the transformer and see if we can find qualitative ingredients of the BP algorithm. And, indeed, we find that all the performed tests point to a similar computational structure as in BP, with successive combinations on increasing block sizes and with ancestry information becoming “available” at the correct layer, to be mixed with the information from the leaves. Importantly, this affinity emerges naturally, without being explicitly enforced at training.
> > >
> > > On the other hand, we don’t fully understand why the reviewer is confident that “a model that solves the tasks in section 3 without BP will also yield the observed patterns”.  For example, a large enough 2-layer network, trained to memorize **all** possible input-output mappings, will not produce the same patterns (and would not implement a similar sequential computation). Instead, if the task is to be solved in a sequential way, we do agree that the same mixing patterns must be employed, and this is in fact what we observe and state in our results.

---

> > > ### Author Response · Authors · 2024-11-27
> > > **Re 2 (2/2):**
> > >
> > > **About C2**
> > >
> > > Our claims on the affinity of transformer and BP computation are based on all the evidence presented throughout the paper: the equivalent input-output mapping, the presence of a compatible mixing pattern, and the sequential emergence of information about the ancestors of a leaf in the hidden representations of the corresponding token.
> > >
> > > On the latter point, following the criticism of the reviewer, we decided to restructure also the writing of the probing experiments paragraph (see final revision), in order to clarify the weight of the evidence. We also repeated the experiments with more training samples for the probe (2^14) to achieve better accuracy and a cleaner plot. Since the available space in the main is limited, we moved the right panel to the appendix, and reserved some additional space for the following explanations:
> > >
> > > *“Keeping the encoder weights frozen, we investigate how much information about the ancestors of any leaf is contained in the successive hidden representations of the
> > > corresponding token. While in the exact embedding of BP the k-th level ancestor information must be available at layer k to iterate the recursion for the downgoing messages, the MLM training does not set such a requirement. To probe the encodings, we employ a special-
> > > ized two-layer readout for each encoder-layer/ancestry-level pair— independent of the token position—trained on a supervised dataset with 214 examples. In Fig. 7, we show that the prediction accuracy is high on ancestors up to the same level as the probed layer and deteri-
> > > orates on higher levels of ancestry. Note that, unless the information about the entire block of 2ℓ−k tokens is properly mixed in through the attention mechanism, a perfectly accurate prediction of the common kth level ancestor from a single token representation is impossible, as the mapping becomes non-deterministic. Moreover, the “overfitting” scenario, where the ancestors are reconstructed solely by the trained probes and the sequential reconstruction is an artifact, can be ruled out by considering the gap between the accuracies achieved from different layers—the relative comparisons are fair since the readouts are trained on the same datasets—, and by training the probes only on some positions—see Appendix D.6”*
> > >
> > > What was not clear from our previous writing, but is important to appreciate the result is that:
> > > * The 1-hidden layer probes (hidden dim=64) are attached to single token embeddings—i.e., **no recombination can be performed**.
> > > * The probes are trained with the same data and for the same amount of epochs between different layers—i.e., the **probes should not overfit more for one layer than the others**.
> > > * We use the same readout for all positions, and in the appendix show that it needs not to be trained on all positions to retain effectiveness across positions —i.e., **position dependence of the probability distributions for the symbols cannot explain the sequential effect**.
> > >
> > > We think these experiments do support the thesis that some trace of the BP computation can be found in the transformer activations.
> > >
> > > We thank the reviewer again for their efforts and hope our work during this discussion phase can convince them to raise their score.

---

> > > > ### Comment · Reviewer_c8Kw · 2024-12-03
> > > >
> > > > I thank the authors again for additional clarifications about the claims being made and adding important details on the probing experiments in the updated version of the paper.
> > > >
> > > > C1: I understand the authors have claims both about computational equivalence as well as some level of algorithmic equivalence between the learned transformer and BP, in the paper as a whole. I also agree with the authors that evidence in section 3 supports claims on computational equivalence. Nonetheless, alternatives to the BP algorithm exist. Take MLM as an example, which is the task focused on by the probing experiments. The authors state that a wide enough two-layer net could memorize input-output mapping but wouldn’t match the BP marginals. Why must this be the case? The training objective is MLE (log-loss), on samples drawn from the true distribution. As we increase the amount of training samples for the two-layer net, wouldn’t it eventually match the ground truth conditional distribution of the masked token given observed tokens, and thus match BP marginals, assuming the two-layer net is expressive enough?
> > > >
> > > > Suppose by argument of scarcity of training data this alternative is ruled out, there exist other alternatives still - running BP on factor graphs that have equivalent distributions but different graph compared to the ground truth binary tree structured factor graph. You can obtain such factor graphs by combining two or more small factors in the ground truth graph into a single larger factor, and/or by marginalizing out latent variables from the ground truth graph. At the extreme is a factor graph that only has a single factor which is connected to all sixteen observed variables, and which essentially just encodes the joint distribution with a single factor. Approximations to these solutions would behave similar to approximations to BP on the test data because they approximate the same ground truth distribution over $x_{1:16}$.
> > > >
> > > > C2: Thank you for adding details on probing experiments. Given the evidence I agree with the author that the frozen encoder representations of a token at each layer contain increasing amounts of information about the $2^{l-k}$  block, and this information is useful to predicting the ancestors one layer up. However, a question is whether this information is simply the assignment to the corresponding block of variables (e.g. at the first layer the observed size-2 blocks, the second layer size-4 blocks, etc), or whether it is some some deeper embedding of it (e.g. the up- or downward messages of running BP given the observed values of the corresponding blocks). The strength of the evidence varies across the layers.
> > > >
> > > > From a difficulty perspective, learning to predict the upper layer variables (e.g. the root) using a two-layer readout that takes shallow embeddings of $x_{1:16}$ is arguably very difficult (since, in figure 3, even a four layer transformer takes more than $2^{14}$ sequences, the number of sequences used in probing, to become perfect at the root prediction task), and it is also more difficult than predicting lower layer variables from their smaller corresponding blocks. So the evidence for higher layer transformer encodings capturing easily decodable features of root, and not just $x_{1:16}$ seems strong. However, for lower level such as predicting the ancestor of a block of size 2 or 4, I wouldn’t be surprised if shallow concatenated word embeddings encoding the identity of the block (instead of the corresponding transformer encodings) is enough as input to a two-layer readout to learn to perfectly predict their ancestor, because while $2^{14}$ is not enough for root prediction of the entire block, it may very well be for predicting roots of smaller blocks. Thus it is actually surprising that the probe doesn’t do ~100% in figure 7 on predicting the level 3 ancestor using the first layer representation (Fig3, triangle@3). Could it be doing something different from pooling information from immediate neighbors (which is what BP would do)?
> > > >
> > > > Overall, while the paper presents several pieces of evidence that is compatible with a transformer that implements BP, I find the current evidence still insufficient to conclude whether 1) the transformer is approximating BP on the ground truth tree graph or doing something in-between BP and brute-force and 2) whether the observed behavior (e.g. marginal match / attention pattern / probing trends)  in this paper would also arise on more complicated graphs, or even the same graph but with different transition matrices that allow for ambiguity. The particular structure of the transition matrix used in this study (for all levels of filtering) has the property that knowing both left and right child uniquely determines parent, and so many of the upward BP messages would effectively be computing this deterministic mapping from pair of children to parent (up to the filtered level), which is a very special case of BP messages.

---

> > > > > ### Author Response · Authors · 2024-12-03
> > > > > **Re 3 (1/2):**
> > > > >
> > > > > ```Alternatives to the BP algorithm exist…take MLM…authors state that a wide enough two-layer net could memorize input-output mapping but wouldn’t match the BP marginals. Why must this be the case? Training objective is MLE (log-loss), eventually match the ground truth conditional distribution, assuming the two-layer net is expressive enough? Suppose by argument of scarcity of training data this alternative is ruled out```
> > > > >
> > > > > We introduced the wide two-layer thought experiment to challenge the statement of the reviewer: “a model that solves the tasks in section 3 without BP will also yield the observed patterns”. We assume the reviewer now agrees with us that this is not the case.
> > > > >
> > > > > On their new remark: a sufficiently wide two-layer network is a universal approximator because, in the limit, each hidden unit can focus on a specific input and map it with the second layer to the correct output, and not because any function can be exactly rewritten in terms of a (linear operation + non-linearity + linear operation).
> > > > >
> > > > > In fact, with a single non-linearity, it is not possible for the network to perform all the aggregations that are needed in the proposed hierarchical model, so only a pure memorization approach could lead the model to fitting the training data (e.g. with a piece-wise linear function if the non-linearity is ReLU). When new inputs are presented, the model would at best produce a linear interpolation between the closest inputs in the training set, which does not yield a good predictor in our data model (note that changing a single symbol in the sequence can lead to a completely distinct set of ancestors). Therefore, as conceded by the reviewer, in the data-scarce regime we are considering, the good calibration on new examples, in- or out-of-sample, rules out a pure memorization strategy.
> > > > >
> > > > >
> > > > > ``` There exist other alternatives still - running BP on factor graphs that have equivalent distributions but different graph compared to the ground truth binary tree structured factor graph…combining two or more small factors in the ground truth graph into a single larger factor…or by marginalizing out latent variables…approximations to these solutions would behave similar to approximations to BP.```
> > > > >
> > > > > The BP derivation does not introduce additional assumptions on the underlying data distribution, it is just an exact way of enforcing the correct relations between the hidden and visible variables, deriving from the top-down Markovian nature of the generative process. The alternative factor graph representations proposed by the reviewer are completely equivalent to the original BP graph, since the messages reaching the leaves will still need to be computed in the same way (the more complicated factors will still impose the true underlying hierarchical correlation structure). Therefore, we agree that all approximations of these algorithms are equivalent.
> > > > >
> > > > > We don’t understand how this point contradicts any of the claims we make in the paper (see also Re 2 (1/2)). Moreover, the observed token mixing patterns (through the attention mechanism) seem to point to a rather transparent interpretation of how correlation lengths are progressively accounted for by the transformer, which is in itself a significant mechanistic interpretation contribution.
> > > > >
> > > > >
> > > > > ```However, a question is whether this information is simply the assignment to the corresponding block of variables…or whether it is some deeper embedding of it. The strength of the evidence varies across the layers. Difficulty perspective: to predict the upper layer variables (e.g. the root) using a two-layer readout that takes shallow embeddings of x1:16 is arguably very difficult, more difficult than predicting lower layer variables from their smaller corresponding blocks. For lower level…shallow concatenated word embeddings…is enough.```
> > > > >
> > > > > We agree with the reviewer that approximating the ancestor computation for a single level might be easy with a two-layer readout once the 2-blocks are identified, and that achieving such good approximation for higher ancestors becomes increasingly difficult (likely impossible if the size of the hidden layer is kept fixed to 64). Precisely for this reason, since higher ancestor classification (including the root) is in fact achieved by the same simple read-out from the higher encoding layers, it is clear that some steps in the computation need to have taken place in the previous layers. A simple linear mixing within larger and larger blocks would not suffice, since the whole BP computation would need to be approximated by the read-out, and the reviewer agreed this would be implausible.
> > > > >
> > > > > Moreover, this discussion focuses only on the ancestor prediction, disregarding that this computation is auxiliary for the exact MLM inference but spontaneously appears in the transformer.

---

> > > > > ### Author Response · Authors · 2024-12-03
> > > > > **Re 3 (2/2):**
> > > > >
> > > > > ```Thus it is actually surprising that the probe doesn’t do ~100% in figure 7 on predicting the level 3 ancestor using the first layer representation (Fig3, triangle@3). Could it be doing something different from pooling information from immediate neighbors (which is what BP would do)?```
> > > > >
> > > > > We agree that this observation is somewhat surprising, at first, since it shows that the embedded computation does not follow exactly the “BP order”, however, we argue it could have been predicted from the attention maps.  If one looks closely at the first layer map in the top row (i.e. the first attention layer), the attention pattern is not exactly the one we imposed in our transformer BP construction, and does not follow exactly the 2-block structure (although it does focus on this range of correlations). In order to obtain a good approximation of BP, some parts of the “missing” recombinations must have been postponed to the following transformer layers. Some brute-force memorization of mappings between blocks of symbols and their common ancestors might have taken place, but without a supervised signal. We find it somewhat remarkable that such a discrepancy only clearly occurs in the first layer.
> > > > >
> > > > > This observation is one of the reasons we do not argue that the computational steps of the transformer are in 1-to-1 correspondence with BP. Yet, traces of the BP computations can be found in the trained transformer. In the paper, we only claim a compatibility/affinity between the sequential structure.
> > > > >
> > > > >
> > > > > ```Current evidence still insufficient to conclude 1)```
> > > > >
> > > > > Both interpretations proposed by the reviewer entail an affinity of computation between BP and the transformer, which is what we claim. Even if the transformer memorizes a table to impute the ancestors from blocks of larger sizes, it is still a sign that it is memorizing a piece of the BP computation, and not some specific input-output mapping.
> > > > >
> > > > > ```Current evidence still insufficient to conclude 2)```
> > > > >
> > > > > We somewhat agree that our understanding of what the transformer implements cannot be straightforwardly generalized to other, more complicated graphical models. However, we would like to remind them that i) this is not something that we claim, in fact we explicitly state in our revised conclusion that carrying out similar experiments on different graphical models would be very instructive; ii) our work has to be taken in the context of current state-of-the-art mechanistic interpretation studies, which in most cases focus on very narrow e.g. arithmetic problems.
> > > > >
> > > > > On the reviewer’s new remark about the non-ambiguous nature of the model: note that the structure of our transition matrices indeed rules out ambiguity going up the tree, but not for the MLM task which requires the descending messages, as demonstrated by the fact that the exact marginals from BP are not delta-like. As can be seen from the BP perspective, there is no significant increase in difficulty for the MLM task in the case of child-to-parent ambiguity, since the same computation has to be carried out. Moreover, our exact transformer implementation of BP applies to any transition tensor. Finally, we would have been happy to show some results in an ambiguous case if this point had been raised before, but now we can no longer revise our paper.
> > > > >
> > > > >
> > > > > **Final remarks**
> > > > >
> > > > >
> > > > > We encourage the reviewer to revisit the weaknesses they first identified in our paper, and how we addressed them throughout this rebuttal period. Namely:
> > > > >
> > > > > ```1) It’s not clear how much the observations made in this work generalizes to larger models, and more complicated data distributions.```
> > > > > We agree with this point, but as stated this is a limitation that any mechanistic interpretation study will necessarily suffer from.
> > > > >
> > > > > ```2) The empirical evidence presented has alternative interpretations that haven't been ruled out.```
> > > > > Here, we believe that we have added stronger evidence, better explanations and more complete discussions. This improvement has been recognized by the reviewer in our exchanges.
> > > > >
> > > > > ```3) The construction for implementing BP on depth- tree using a transformer of only -layers could benefit from a bit more details, especially on the root-to-leaf message passing part.```
> > > > > This was addressed in our first updated draft, to which the reviewer responded very positively.
> > > > >
> > > > > Overall, we tried to take into account all the received feedback. We think our paper strongly benefitted from this, and we sincerely thank the reviewer for their engagement. However, at this point, we think it would also be fair for the reviewer’s score to reflect our efforts and the paper improvements since the original version was judged to be only marginally below threshold.

---

> ### Author Response · Authors · 2024-11-25
> **Re:**
>
> We thank the reviewer for engaging in the discussion. We would like the reviewer to clarify their point in concern C1, if possible, before we try to answer their further questions. It seems that the reviewer is stating that the transformer can:
> * not only solve the tasks in section 3
> * but obtain the same input-output mappings as BP, both on the training distribution and out-of-sample (even on random inputs)
> but with a completely different, unrelated computation.
>
> Is there some known example of a similar scenario, where two unrelated non-linear functions end up producing the same continuous outputs on all inputs, without this match being enforced explicitly?
>
> In our understanding, for example, in standard situations if you train two neural networks on the same data, in the end they might align their outputs on the training data distribution. However, they will still provide different outputs on random data.
>
> Is the reviewer suggesting there exists a completely different way of performing exact inference on a tree? And more generally, what evidence would be sufficient to support the claim that an architecture implements algorithm X?
>
> We thank again the reviewer, we will follow up and address the technical points raised above.

---

### Author Response · Authors · 2024-11-19
**Response to All Reviewers**

We thank all the reviewers for their thoughtful and helpful feedback! We are pleased that:

**Reviewer c8Kw** finds that our work is “well-placed” in the context of mechanistic interpretability of transformers, as well as in the study of the “effect of structured data on machine learning models”.

**Reviewer VwvV** finds our analysis of the expected attention patterns, and the observation of sample efficiency gain via MLM pretraining “quite interesting”.

**Reviewer BYbw** finds that our work offers a “novel viewpoint to study transformer learning from belief propagation” and that the “CFG construction with filtering is interesting”.

More importantly, we thank the reviewers for carefully identifying some significant weaknesses and avenues for improvement, which pushed us to perform new experiments and undergo a significant rewriting of the paper to deliver our points in a clearer fashion. We are currently working on the revised manuscript, and will upload it as soon as possible.

In the meantime, we wanted to address the weaknesses and questions raised by each reviewer in individual responses below. Please do not hesitate to let us know if you have additional comments or questions, which will allow us to achieve the best possible version of our paper.

---

### Author Response · Authors · 2024-11-23
**Revised version**

Following the comments and suggestions of the reviewers, we have extensively revised our presentation and included additional experiments and discussions in our paper. We believe the clarity of the paper has substantially improved, and for this, we thank the reviewers for their constructive feedback.

Here's a list of the main changes you will find in the uploaded revised version:
* The abstract was modified to better reflect the goals of our work.
* An **Our contributions** paragraph was added to the introduction, listing the main results and clarifying the novelty and the message of the paper.
* The main figure, Fig. 1, was reorganized to include stronger evidence of the computational equivalence between the trained transformer and the exact algorithm, and the sequential learning process in the implementation of the hierarchical correlations in the data.
* The section on the data model, **A model with filtered hierarchical correlations** was shortened and simplified, moving the technical descriptions that are not necessary for understanding the main results to the Appendix. Importantly, we now more clearly state the purpose of considering a simpler setting than standard CFGs.
* The main findings, substantially rewritten to improve clarity, are now described in two separate sections:
     - **How transformers learn to climb the hierarchy in time**: Here, we show evidence that in the root classification and in the MLM tasks, transformers not only approach optimal accuracy but approximate the output of the exact oracle. Moreover, we show that during the learning process, the model integrates higher hierarchical levels sequentially, leading to a staircase behavior.
     - **How transformers embed the exact inference computation**: Here, we show that when the number of transformer layers is matched to the levels in the generative tree, the computation can become interpretable. We present our feasible implementation of BP, show the analysis of the attention maps, the probing experiments at the different encoder levels, and the MLM pre-training effect.
* In the **Conclusions**, we now explain how our study could represent an important step in interpreting transformer computation in related settings.
* We included the suggested references and the citations mentioned in the answers to the reviewers.
* We completed and clarified the explanation of the feasible BP implementation within the transformer architecture.

We hope our attempt at improving our work meets the concerns of the reviewers, and that the scores can be raised accordingly.
Please feel free to let us know if there are any remaining comments not addressed. We appreciate any feedback, and we are happy to answer any further questions and adjust the manuscript.

---

### Meta-Review · Area_Chair_Xhyf · 2024-12-04

**Metareview:**

This paper trains Transformer models on a synthetic PCFG in order to compare their mechanisms with an exact solution through belief propagation. Their analysis demonstrates that the attention patterns behave hierarchically in an interpretable way when the layers match the actual tree depth, which matches the behavior of the exact solution. They also show that this hierarchical structure emerges gradually during training.

Although most reviewers found some merit in the various results and appreciated many of the visualizations, overall, several criticisms remained. In particular, all reviewers remain skeptical of the claim that the BP algorithm is the *only* implementation compatible with their results; further confirmation is needed. Broadly, the authors map several of their visualizations to specific claims about the algorithm implemented by the model and they show that the algorithm can be computed by the Transformer model, but while their findings are compatible with these proposed algorithms, they are not strong evidence. Some reviewers are also skeptical of the generality of their findings, given the small models trained on a synthetic setting, although this complaint can be applied to most empirically rigorous work on the science of deep learning.

**Additional Comments On Reviewer Discussion:**

Authors responded adequately to the requests for presentation improvement by clarifying their contributions in the paper and expanding their discussion of specific technical aspects such as the feasibility of a belief propagation implementation within the model. They clarified a number of initial points of confusion on the part of each reviewer, and it is very possible that future revisions of this paper will be treated better because of the clarifications and additional detail in their revision.

The authors also added several experiments that continued to backup the claim that this trained model could be implementing BP, but none of these experiments guaranteed that this was the only possible algorithm.

---

### Decision · Program_Chairs · 2025-01-22

Reject